# Elucidation of Spatiotemporal structures from high-resolution blowing snow observations

Kouichi Nishimura[1], Masaki Nemoto[2], Yoichi Ito[2], Satoru Omiya[3], Kou Shimoyama[4], Hirofumi Niiya[5]

[1]Nagoya University, Nagoya, 464-8601, Japan
[2]Snow and Ice Research Center, NIED, Shinjo, 996-0091, Japan
[3]Civil Engineering Research Institute for Cold Region, PWRI, Sapporo, 996-0091, Japan
[4]Institute of Low Temperature Science, Hokkaido University, Sapporo, 060-0819, Japan
[5]Research Institute for Natural Hazards and Disaster Recovery, Niigata University, Niigata, 950-2181, Japan

*Correspondence to*: Kouichi Nishimura (knishi99@gmail.com)

**Abstract.** Systematic observations were conducted to investigate the spatio-temporal structures of blowing snow. Along a line perpendicular to the dominant wind direction on the leeside of a flat field, fifteen Snow Particle Counters (SPCs) and Ultra Sonic Anemometers (USAs) were placed 1.5 m apart. Data were recorded at high frequencies of 100 kHz for SPCs and 1 kHz for USAs. The horizontal mass flux distributions, representing the spatio-temporal variability of blowing snow, exhibited non-uniformity in both time and space and manifested periodic changes akin to snow waves. Additionally, the presence of 'snow snakes,' meandering near the snow surface, was observed. Quadrant analysis revealed predominant snow fluxes in quadrants Q1 ($u'>0$, $w'>0$) and Q4 ($u'>0$, $w'<0$). However, a more detailed parametric curve analysis indicated the existence of ejection events Q2 ($u'<0$, $w'>0$) before snow waves and in front of snow snakes, shifting to Q1 and Q4 afterward, implying the consideration of both top-down and bottom-up mechanisms for burst sweep events.

## 1 Introduction

The transport of snow by wind, known as blowing or drifting snow, plays a significant role in engineering and climatological contexts. Near roads, blowing snow can cause snow drifts and reduce visibility, leading to traffic accidents. Wind-driven snow redistribution can result in localized snow accumulation and the formation of snow cornices, influencing snow avalanche danger. The impact of katabatic winds on the Antarctic ice sheet affects mass and energy balance, becoming more critical when considering global climate change effects.

Numerous investigations have been carried out, both in the field and in wind tunnels, to explore the meteorological conditions that trigger blowing snow and to gain insights into its internal structures. However, the model of snow transport is divided into two main categories: Lagrangian models, which focus on the motion of individual snow particles in the saltation layer near the snow surface, and turbulence diffusion models, which deal with the suspension layer. Notably, Nemoto and Nishimura (2001) developed a random flight model of blowing snow that incorporates turbulence, inertia of snow particles, and particle size

distribution. This model successfully captures the entire transition from the saltation layer to the suspension layer during a blizzard. However, it is limited to steady-state conditions and does not address temporal and spatial variations.

To gain a more comprehensive understanding, it is crucial to consider organizational structures such as turbulence sweeps and ejections, and intermittent dynamics and non-linear dependence on wind speeds, when discussing the onset and development of snow transport. Unfortunately, the existing time-averaged models based on local equilibrium theory and isotropic turbulence

theory have limitations in grasping coherent structures with regularity due to their restricted flow field representation.

Moreover, it has been observed that traffic accidents caused by reduced visibility on roads are strongly influenced by temporal and spatial variations rather than the intensity of blowing snow itself (Kajiya et al., 2001). Consequently, the demand for models capable of describing these temporal and spatial structures is widely recognized. Recently, efforts have been made to develop models that combine Large-Eddy Simulation (LES) with Lagrangian snow transport models (e.g., Groot Zwaaftink et

al., 2014 and Okaze et al., 2018). However, these models lack comprehensive measurements that adequately address the high variability of these phenomena.

While a study by Aksamit and Pomeroy (2016, 2017) focused on temporal changes, it unfortunately lacked spatial information. As a result, the reliability of these models remains unconfirmed, and our understanding of the high-frequency fluctuations in the internal structures of blowing and drifting snow remains unsatisfactory. Against this backdrop, our study aims to

systematically measure blowing and drifting snow to investigate their spatiotemporal structures. To achieve this goal, we have deployed fifteen Snow Particle Counters (SPCs) in designated test areas and conducted measurements using an equal number of ultrasonic anemometers, providing high temporal resolution data. While the dataset obtained from our series of observations is indeed substantial, this study primarily offers a brief overview of these observations and the results obtained.

## 2 Field observation

The field observations were conducted during the winter of 2018 in Tobetsu, Hokkaido, Japan. Tobetsu is located close to the coast of Ishikari bay and experiences strong monsoon winds from the west during the winter. The measurement apparatus zone was situated in the windward area, where a flat snow field spreads over a distance of more than 200 meters (Fig. 1).

Fifteen measurement towers were installed along a line perpendicular to the dominant wind direction, with each tower spaced 1.5 meters apart. The towers were equipped with Snow Particle Counters (SPCs, SPC-7 and SPC 95, Niigata Denki Co. Ltd)

and Ultra Sonic Anemometers (USAs, Young 81000) to measure both the spatio-temporal structures of blowing snow and the air flow (Fig. 1). The height of the SPCs was initially 39 cm, and the height of the USAs was 84 cm, but both heights decreased with increasing snow accumulation.

The SPC used in this study (Niigata Denki Co.) is an optical device (Nishimura and Nemoto, 2005) that measures the diameter and the number of drifting snow particles by detecting their shadows on a photodiode (assuming that drifting particles are

spherical in shape). Electric pulse signals resulting from snow particles passing through the sampling volume ($2 \times 25 \times 0.5$ mm) are sent to a transducer and an analyzing data logging system (PC). In this way, the SPC is able to detect particles in the range of 40–500 μm. The analysis software divides the particles into 64 size classes and records the number of particles in

each size class at 1-s intervals. The SPC is mounted on a self-steering wind vane, and hence the sampling region, which has a cross-sectional area of 2 × 25 mm (50 mm$^2$), is maintained perpendicular to the horizontal wind vector. If the diameter of a

snow particle is larger than that of the maximum diameter class, the snow particle is considered to belong to the maximum diameter class. Usually, SPCs are used to obtain snow particle size distribution and mass flux at 1-second intervals. However, in this study, the output signals from the SPC transducers were directly recorded with a high frequency of 100 kHz. This allowed the researchers to calculate not only the mass flux but also the particle size and speed with fine time resolutions (Nishimura et al., 2014). Interestingly, the SPC has also found application in sand transport research as a "Sand Particle

Counter" (Yamada et al., 2002 and Mikami et al., 2005). All SPC sensors are accurately calibrated in advance using spinning wires of various diameters, enabling us to effectively account for any sensitivity differences. Detailed procedures are provided in Sato et al. (1993). Further, over the snow surface, we can generally avoid the effect of the dust and fine soil contamination over the optical parts, unlike the situations over sand and soil surfaces. Numerous long-term observations have been conducted in the Alps (e.g., Naaim-Bouvet et al., 2010, 2014; Gilbert, 2019), the Arctic (e.g., Lenaerts et al., 2014; Frey et al., 2020), and

Antarctica (e.g., Sigmond et al., 2021; Wever et al., 2023), all of which attest to the reliability of the SPC. Generally, we can leave the system even for the whole winter without cleaning or wiping the optical parts. Particularly this campaign was carried out only for less than two months. Thus, we believe the contamination over the optics can be reasonably neglected.

The wind speeds measured with USAs were stored with a frequency of 1 kHz. To ensure accurate synchronization of time for

both recordings, precise calibration and alignment were performed. As elucidated further in subsequent sections, the heights of the sensors from the surface fluctuated due to snow accumulation. Consequently, we took the necessary step of manually measuring the sensor heights of both the Snow Particle Counters (SPCs) and Ultrasonic Anemometers (USAs) at every observation period. This meticulous approach ensured that any variations in sensor height were accurately accounted for, maintaining the integrity and reliability of our measurements throughout the experiment.


Other meteorological data, such as mean wind speeds and directions, and air temperature, were obtained from an Automatic Weather Station (AWS) located approximately 1 km away in the northwest (windward direction). Additionally, data on precise precipitation were available from the Double Fence Intercomparison Reference (DFIR) set at the Ishikari Blowing-Snow Test Field by Civil Engineering Research Institute for Cold Region (CERI), located at a distance of nearly 4 km to the west. The

DFIR data were primarily used to determine whether the observed phenomena were purely drifting or blowing snow or if they were accompanied by snowfall.

Overall, the field observations were conducted in a well-equipped measurement zone, allowing the researchers to collect detailed data on blowing snow and atmospheric conditions, which are crucial for their study of spatio-temporal structures of blowing snow.

## 3. Results and analysis

In February 2018, ten minutes average of the meteorological conditions were recorded using AWS and DFIR, as depicted in Fig. 2. The red arrows indicate the duration of the intense observation period. Despite not having an exceptionally high wind speed at 2 meters above the surface, the westerly wind was sufficient to initiate drifting/blowing snow. Throughout the period, the air temperature remained mostly below zero, causing the snow depth to increase from 80 to 110 cm in just one month. Consequently, the sensor height of the Snow Particle Counter (SPC) decreased from 39 to 1 cm due to the snow accumulation. During the intense observation period, the Richardson number ($Ri$), calculated at the AWS site, using vertical temperature and wind speed profiles (Sutton, 1953), indicated neutral conditions ($Ri < |0.01|$).

Figure 3 illustrates the recordings of wind speed from the ultrasonic anemometer (USA) and snow mass flux, particle speed, and particle diameter obtained with the SPC at the spanwise direction Y = 9 m for 60 seconds. On February 16, the sensor heights for the USA and SPC were 91 cm and 23 cm, respectively, while on February 24, they were 76 cm and 8 cm. Although the sensor height of the USA is 70 cm higher than the SPC, the change in snow mass flux aligned well with the wind speed on both days. Particularly, the response on February 24 was more pronounced, suggesting that the snow flux in the saltation layer is more sensitive to wind speed fluctuations than the one in the higher suspension layer. Figure 3 reveals that some particles move rapidly, even surpassing the wind speeds. These particles likely originate from higher positions, where they are accelerated by stronger winds. However, in general, particle speeds are lower than the wind speed, as reported in Nishimura et al. (2014), who found that mean particle speeds are consistently 1 m/s to 2 m/s less than wind speeds at heights of 0.015 to 1 m. Although the sensor height of SPC was lower than that of USA, rough estimates based on the logarithmic wind profile support this trend. While particles smaller than 100 μm in diameter contributed more on February 16, they displayed a wide distribution within the range of 40 – 500 μm on February 24. Decreasing the height from the snow surface revealed larger particle numbers with a wide size distribution.

Figure 4 displays horizontal mass flux distributions, reflecting the spatio-temporal variability of blowing snow, measured with fifteen SPCs for 120 seconds on February 12, 16, and 24. The measured heights from the snow surface were 31 cm, 23 cm, and 8 cm, respectively. At the AWS point, wind direction was kept steadily as west and wind speeds at 2 m high on three days were nearly the same: from 4.5 to 5 m/s. Further, precipitation observed with DFIR was quite small and can be neglected. The three panels in Figure 4 depict maps of snow transport intensity, created by aligning time series data from the Snow Particle Counter (SPC) transverse array in the spanwise Y direction and applying Taylor's frozen hypothesis (1938) to substitute time for the streamwise direction. In the figure, the 'advected distance' is presented on the x-axis, estimated using the average wind speed provided by the ultra-sonic anemometer (USA) and corrected for the height differential between the USA and the Snow Particle Counter (SPC) sensor heights, assuming the logarithmic wind profile.

Despite steady westerly wind direction and similar wind speeds at 2 m height on the three days, the flux distributions were neither uniform with time nor space and exhibited periodic time, in other words, lateral to the streamwise direction, variation. The autocorrelations of horizontal mass flux in Fig. 4 are depicted in Fig. 5. Each line in the figures represents the output from

an individual SPC. While the distributions on Feb. 20 appear somewhat subdued, distinctive peaks are evident on Feb. 12 and Feb. 16, occurring approximately every 10 s, 35 s, 70 s, and 90 s. These correspond to distances of approximately 30 m, 115 m, 230 m, and 300 m, respectively.

Figure 6, which shows pictures of the field's upstream at night on February 20, highlights the periodical snow waves approaching the sensors from back to front. These waves correspond to the periodical change in snow flux observed in Fig. 4. Although the formation mechanism of the snow wave has not been clarified yet Kobayashi (1980) suggested that the snow waves strongly reflect the turbulent flow structure of the wind, and its formation mechanism is similar to the wavelike cloud in the atmosphere and the wind wave in the ocean.

On February 16 and 24, Figure 7 displays the blowing snow flux distributions for 10 seconds at heights of 23 cm, 8 cm, and 1 cm from the snow surface. Due to the substantial increase in snow depth, nearly 10 cm within a single day on Feb. 24, as depicted in Fig. 2, we have introduced two scenarios with SPC heights set at 8 cm and 1 cm. Contrary to Fig. 4, as the height decreased from the surface, long and narrow structures emerged prominently. These structures seem to correspond to the phenomena known as "snow snakes," which meander laterally, merging and bifurcating as they move downwind as shown in Figure 6, similar to observed phenomena in drifting sand movement. The height of the long structures along the snow surface varied, and taller parts occasionally protruded to higher positions, as seen in the cases of 23 cm and 8 cm in Fig. 6. Figure 8 illustrates the 2D-autocorrelation of horizontal mass flux on Feb. 24, depicted at the bottom of Fig. 7. The 1.5-meter spacing between SPCs may not always be sufficient to capture precise structures. However, the Delaunay triangulation (Cheng et al., 2012) used in Figure 7 is a powerful tool for interpolation, mesh generation, and graphical applications. It is widely used in Geographic Information Systems (GIS) to create terrain models, where triangulating elevation points constructs a surface that accurately represents the terrain with minimal distortion. Consequently, the structures observed at 1 cm above the surface, with widths around 2 meters, peak fluxes about 30 cm wide, and lateral spacing of approximately 5 meters, as shown in Figure 7, are quite plausible. Notably, the 2D autocorrelation of horizontal mass flux in Figure 8 indicates a lateral spacing of about 5 meters, which is more than three times the 1.5-meter sensor spacing.

The propagation speed was roughly estimated to be 2 to 3 m/s based on particle speed data from analyzing the SPC data (Fig. 3b).

Figure 9 displays the power spectra of mass flux at Y=10.5 m for February 24 in Fig. 4, along with the ones for wind speeds. The dominant frequencies and general trends of the power spectra for snow flux and wind speed appear very similar, implying that both vary in a correlated manner. Despite the sensor heights of SPCs being 31 cm lower than those of anemometers, these observations suggest that changes in snow flux reflect the structures of turbulence eddies near the snow surface.

We applied quadrant analysis in this study and focused on turbulent sweeps and ejections to better understand the spatiotemporal structures of blowing snow. While previous discussions have centred on the amount of snow transport by wind, often using averaged wind speed over specific time periods, such as the cube to 5th of mean wind speed for ten minutes (e.g., Dyunin, 1967; Mann et al., 2000; Nishimura and Hunt, 2000). However, in order to elucidate the spatiotemporal structures of

blowing snow, it is indispensable to set on our focus on the turbulent structures of the wind. To explore the correlation between snow transport and turbulent coherent structures in the boundary layer, Quadrant analysis (Wallance, 2016) was applied. This classical analysis examines the fluctuating wind velocities of the streamwise and vertical components ($u'$ and $w'$) and their

product, the Reynolds stress, to identify ejection Q2 ($u'<0$, $w'>0$), where lower-momentum air is expelled away from the bed, and sweep Q4 ($u'>0$, $w'<0$), where higher momentum air is pushed towards the bed, events contributing to turbulent energy production. Figure 10 shows the horizontal distributions of snow mass flux, wind speed, and ejection (Q4) and sweep (Q2) events on February 12 and 24, 2018. It is evident that large snow mass flux, corresponding to the ridge of the snow wave (in Fig. 6), occurs under conditions of high wind speeds and sweep events (Q4, $u'>0$, $w'<0$) on both days.

Figure 11 illustrates the correlation between the horizontal snow flux at 8 cm and the absolute value of the kinetic shear stress $u'w'$ in each quadrant on Feb. 24, 2018. The results indicate that snow fluxes were predominantly observed in quadrants Q1 ($u'>0$, $w'>0$) and Q4 ($u'>0$, $w'<0$), while being far less prominent in quadrants Q2 and Q3. Table 1 displays the contributions of each quadrant, acquired on February 12, 16, and 24, at corresponding heights from the snow surface: 31 cm, 16 cm, 8 cm, and 1 cm; analysis was conducted with all the sensor data. Ejection events (Q2: $u'<0$; $w'>0$) and sweep events (Q4: $u'>0$;

$w'<0$) contributed 12% to 22% and 33% to 48%, respectively. Q4 exhibited the largest contribution, while Q1 consistently followed in all events, regardless of the height from the surface. Moreover, the combined contributions of Q1 and Q4 account for over 64%, while Q2 and Q3 contribute less than 36%. The contributions from quadrants Q2 and Q3 ($u'<0$; $w'<0$) were notably smaller compared to the significant impact of Q1 and Q4. Figure 11 suggests a decrease in snow flux with increasing Reynolds stress, which may initially appear contradictory to conventional understanding. To further explore this relationship,

Figure 12 examines the correlation between snow flux and the fluctuating components of u' and w'. Subfigure (a) demonstrates a clear increase in snow flux with u', while Subfigure (b) illustrates a distribution concentrated around zero, resembling a normal distribution. Subfigure (c) reveals a negative trend between u' and w', indicating that w' decreases as u' increases. The relationship between the product of u' and w' and snow flux is depicted in Subfigure (d). Although setting specific hole sizes in Figure 11 may result in variations, Leenders et al. (2005) suggested that vertical fluctuations are inherently constrained by

the distance from the bottom boundary and the overall scale of the structure, leading to poor correlation between surface fluctuations and vertical fluid motions higher in the profile. Consequently, measurements of Reynolds shear stress around 1 m from the surface are typically poorly correlated to snow transport flux.

   In contrast, Fig. 13 illustrates the distributions of the snow mass flux at 1 cm, as well as the wind speed, ejection, and sweep structures at 40 cm for a 10-second on February 24. The data indicates that the foremost region of the snow snake primarily

aligns with the ejection phenomenon, trailed closely by the sweep motion. For a more detailed examination of the structures preceding and during the onset of transport escalation, specific ten-second sub-periods (marked with arrows in Fig. 10 and whole durations in Fig.13) have been extracted and presented in Figs. 14 and 15. Figures 14 (a) to (e) present the wind speeds and mass fluxes at Y=4.5 m, 10.5 m, and 15.0 m from 25 to 35 s and at Y=10.5 m and 15.0 m from 60 to 70 s in Fig. 8. Furthermore, using the approach introduced by Aksamit and Pomeroy (2017), parametric curves of ($u'$(t), $w'$(t)) were displayed,

representing the duration of the time indicated by arrows in each figure: from 30 to 35 s for the former and from 67 to 70 s for the latter. These curves were plotted every one second with different colours.

For example, in Figure 14 (a), no snow transport occurred between 30 to 31s at Y=4.5 m, followed by an abrupt increase in snow flux. Afterward, the mass flux diminished to nearly zero from 31.5s to 32.5s, and then increased again from 33 to 34s. The parametric curve of ($u$'(t), $w$'(t)), encompassing the entire first peak of the horizontal mass flux (pre-increase, increase,

and decrease) between 30 to 33 seconds, was situated within quadrant Q2 (ejection region). However, the second increase occurred in quadrants Q1 and Q4, characterized by stronger horizontal wind speeds. Similarly, at Y=10.5 m in Fig. 14 (b), from 32 to 34 s, the parametric curve mainly remained in Q2, covering the duration before the increase and the flowing period leading to the maximum, and then moved to Q1. In Fig. 14 (c), which shows the case at Y = 15.0 m, the snow flux started to increase at 33 s. The parametric curve was mostly in Q4 one second before. However, at the onset of the increase at 33 s, it

shifted to Q2 and later moved to Q1. Figure 14 (d) illustrates the case at Y=10.5 m from 60s to 65s, where the snow flux started to increase at 68 s and reached its peak at 69.1 s. The parametric curve from 67 to 70 s indicated that both periods, one second before the increase in snow transport and the path to the maximum, were in Q2, and it subsequently moved to Q1 and Q4 where the horizontal wind speed was greater. The snow flux change at Y=15.0 m shown in Fig. 11(e) was almost similar to that at Y=10.5 m. Although the staying period of the parametric curve in Q2 was rather short compared to other cases, it was

on Q2 during both the period just prior to the increase in mass flux and the front part, and then it shifted to Q1 and Q4. Overall, the parametric curve patterns indicated that the ejection part Q2 exists both prior to the snow waves and at the nose of the snow snake, while Q1 and Q4, where horizontal wind speed is higher, follow.

In Fig. 15, structures around the snow snake are displayed, showing wind speed and snow flux for ten seconds at Y=15.0 m and the parametric curve from one to 6 s. The front of the snow snake arrived at the sensor at around 2.5 s, and the snow flux

increased rapidly. During 1 to 2 s, the parametric curve remained in Q1, but it shifted to Q2 just in front of the snake head, then moved to Q1 and Q4. It's worth noting that the curve briefly returned to Q2 before the second mass flux peak. During the flowing period, the curve predominantly stayed in Q1 and Q4, where $u$' is positive. In general, the parametric curve follows the ejection part Q2 in front of the snake head and then shifts to Q1 and Q4, similar to the snow wave behaviour in Figs. 14 (a) to (e).


**4 Conclusions**

We conducted systematic measurements of blowing snow, which allowed us to reveal the spatiotemporal structures of the phenomenon. In comparison, research on sand transport, particularly aeolian streamers, has been explored in studies such as Bauer et al. (1998) and Sterk et al. (1998). Bauer et al. (1998) used a hot-wire anemometer and load-cell sand traps, identifying

three typical transport patterns: streamer families, nested streamers, and clouds with embedded streamers. However, these latter two patterns occurred only at high shear velocities. Comparing their classification with our study's findings in Figs. 4 and 7, the structures observed on February 16 and 24 likely correspond to streamer families and nested streamers. Yet, 'clouds

with embedded streamer' was not found in the snow transport, likely because the wind speeds were relatively low and insufficient to transport large amounts of snow particles. It is worth noting that the periodical changes in snow flux shown in Fig. 4 were not mentioned in Bauer et al. (1998) for the sand transport, possibly because their focus was on phenomena occurring within a short period of five seconds, while the snow waves recognized in Fig. 4, with frequencies less than 0.1 Hz, were observed over 120 seconds.

On the other hand, Sterk et al. (1998) utilized a Gill-type anemometer and acoustic sediment sensing instruments to investigate the relationship between sediment flux and Reynolds shear stress. They reported very low levels of statistical explanation, with correlation coefficients smaller than 0.14, which aligns with our findings, as depicted in Fig. 12(d). Furthermore, Sterk et al. (1998) partitioned instantaneous sediment transport values and sorted them according to wind speed quadrant signatures. They observed that the largest mean saltation fluxes occurred during stress excursions located in Q1 and Q4 quadrants, while excursions in the Q2 and Q3 quadrants were unable to sustain appreciable saltation activity. This observation is consistent with the findings of Shonfeldt and von Lowis (2003), Leenders et al. (2005), and Wiggs and Weaver (2012) who employed sonic anemometers for sand transport observations. Our study corroborates these findings, as summarized in Table 1, where contributions from Q1 and Q4 quadrants were significantly larger than those from Q2 and Q3. This supports the notion that horizontal velocity fluctuations ($u'$) play a pivotal role in aeolian sediment transport compared to vertical fluctuations ($w'$) or kinematic shear stress ($u'w'$).

Baas and Sherman (2005) applied the Variable Internal Time Averaging method (VITA) to wind speed data, which identifies significant turbulent events using local variance within a moving time window (Blackwelder and Kaplan, 1976). They found that VITA events were generally poorly correlated with transport events, and sweep-like motions with strongly positive $u'$ signals dominated over ejection-like motions. Then, they interpreted the streamer patterns have a probable origin in large 'eddy surface layer', where turbulent eddies from higher regions of the atmospheric boundary layer travel downward, elongate and stretch out in the shear layer, and scrape across the surface. Later, Hunt and Morrison (2000) proposed a top-down model of boundary layer turbulence for very high Reynolds number flows. Aksamit and Pomeroy (2017) also applied quadrant analysis and VITA techniques to their snow transport measurements, revealing that sweeps were the dominant motion for initiating blowing snow and increasing concentration and particle number flux near the surface. Active ejections occurred during active blowing snow and after sweeps.

Our measurements also represents that the snow fluxes were observed largely in Q1 ($u'>0$, $w'>0$) and Q4 ($u'>0$, $w'<0$) quadrants. However, we should note that the detailed analysis with the parametric curve revealed that the ejection part Q2 exists both prior to the snow waves (in Figs. 14(a) to (e)) and in front of the snake head (in Fig. 15), and then shifts to the Q1 and Q4, where horizontal wind speed becomes larger. Thus, in addition to the top-down mechanism proposed by Baas and Sherman (2005), bottom-up mechanisms of burst sweep events cannot be ruled out at this stage. Indeed, conducting detailed measurements with SPCs and USAs positioned at the same height from the surface would provide more accurate and directly comparable data. Additionally, incorporating the latest turbulence analysis techniques could further enhance our understanding and facilitate reaching conclusive findings. By leveraging advanced methodologies and instrumentation, such as high-

resolution SPC arrays and state-of-the-art turbulence analysis algorithms, we can obtain more precise measurements and delve deeper into the complex interactions between blowing snow and turbulent flow dynamics. These efforts would contribute significantly to advancing our knowledge in this field and ultimately lead to more comprehensive conclusions.

As stated in the introduction, time-averaged blowing snow models have limitations, necessitating the incorporation of relationships with organizational structures such as turbulence sweeps and ejections to discuss the onset and development of snowstorms more accurately. Models that can describe temporal and spatial structures are crucial. To address this, we have developed the Large-Eddy Simulation coupled with Lagrangian Snow Transport model LLAST (Okaze et al., 2020). This model combines large-eddy simulation for turbulent flow and particle translational simulations based on Newton's equations

of motion. The dataset obtained from the series of observations in this study is substantial, providing ample opportunities to derive concrete conclusions elucidating the spatiotemporal structures of blowing snow. However, it's important to note that our analysis is still quite limited at this stage, and only preliminary results introduced here. Nevertheless, more detailed analyses, which will be presented in subsequent manuscripts, hold the promise of significantly contributing to the improvement and validation of the model.


*Data availability*

All raw data can be provided by the corresponding authors upon request.

*Author contributions.*

KN and MN planned the project; KN and MN, YI, SO, KS, and HN took part in the observations; KN and MN analyzed the data; KN wrote the manuscript.

*Competing interests.* The contact author has declared that neither of authors has any competing interests.


*Acknowledgements.*

The authors extend their heartfelt gratitude to the Tobetsu local government for their generous permission to utilize the managed park area as the observation field. Additionally, our sincere appreciation goes out to all our colleagues for their invaluable support in conducting the field measurements. This collaborative effort played a pivotal role in successfully

completing this research endeavour. We also deeply thank the editor and two reviewers for their valuable and constructive comments.

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

**Table 1. Contributions of each quadrant on February 12, 16, and 24, at corresponding heights of mass flux from the**
**snow surface: 31 cm, 16 cm, 8 cm, and 1 cm.**

**Figure 1. Observation field: Tobetsu, Hokkaido, Japan (left). Measurement apparatus: fifteen Snow Particle Counters (SPCs) and Ultra Sonic Anemometers (USAs) set on towers 1.5 m apart (right).**

**Figure 2. Meteorological conditions at the observation site in February 2018. Red arrows indicate the duration of the intense observation period.**

**Figure 3. Time series of wind speed, snow mass flux, particle speed, and particle diameter at the spanwise direction Y = 9 m. a: on February 16, the sensor heights for the USA and SPC were 91 cm and 23 cm, b: on February 24, 76 cm**
**and 8 cm.**

**Figure 4.** Horizontal mass flux distributions for 120 sec., reflecting the spatio-temporal variability of blowing snow, on February 12, 16, and 24. The measured heights from the snow surface were 31 cm, 23 cm, and 8 cm, respectively. Advected distance is also shown on the top x-axis.


**Figure 5.** Autocorrelations of horizontal mass flux shown in Fig. 4. Each line indicates the output from the individual SPC.

**Figure 6.** 'Snow waves' appeared on the field's upstream at night on February 20 (above) and 'snow snake' (below). In both cases the wind blew from back to front. As shown in figures, the snow waves are organized in a lateral or spanwise orientation and the snow streamers are quasi-parallel to the streamwise direction.

**Figure 7.** Horizontal mass flux distributions for 10 seconds at heights of 23 cm, 8 cm, and 1 cm from the snow surface. Advected distance is also shown on the top x-axis.


**Figure 8.** 2D-autocorrelation of horizontal mass flux on Feb. 24 shown in the bottom of Fig. 7.

**Figure 9.** Power spectra of mass flux and wind speeds at Y=10.5 m on February 12, 16, and 24

**Figure 10.** Horizontal distributions of snow mass flux, wind speed, and ejection (Q4:red) and sweep (Q2: blue) events on February 12 and 24, 2018.

**Figure 11.** Correlation between the horizontal snow flux at 1 cm and the absolute value of the kinetic shear stress *u'w'* in each quadrant on Feb. 24, 2018.


**Figure 12.** Relations between the snow flux, the fluctuating component of u' and w', and the product of u' and w' on Feb. 24, 2018. All the data and the observation period are the same as Figure 11.

**Figure 13.** Distributions of the snow mass flux at 1 cm, as well as the wind speed, ejection, and sweep structures at 40 420 cm for a 10-second on February 24.

**Figure 14.** (a) to (e) present the wind speeds and mass fluxes at Y=4.5 m, 10.5 m, and 15.0 m from 25 to 35 s and at Y=10.5 m and 15.0 m from 60 to 70 s in Fig. 8. Furthermore, parametric curves of (*u'*(t), *w'*(t)) were plotted every one

second with different colours for the period indicated by arrows in each figure: from 30 to 35 s for the former and from
67 to 70 s for the latter. The blue arrows indicate the onset of the snow flux increase.

**Figure 15. Structures around the snow snake: wind speed, snow flux for ten seconds at Y=15.0 m, and the parametric curve from one to 5 s in Fig. 10. The blue arrow indicates the onset of the snow flux increase.**





**Table 1. Contributions of each quadrant on February 12, 16, and 24, at corresponding heights of mass flux from the snow surface: 31 cm, 16 cm, 8 cm, and 1 cm.**

| Date<br>Sensor height | Feb. 12<br>16 cm | Feb. 16<br>23 cm | Feb. 24<br>8 cm | Feb. 24<br>1 cm |
|---|---|---|---|---|
| Q1 | 0.30 | 0.30 | 0.33 | 0.28 |
| Q2 | 0.22 | 0.23 | 0.12 | 0.21 |
| Q3 | 0.12 | 0.14 | 0.07 | 0.15 |
| Q4 | 0.36 | 0.33 | 0.48 | 0.36 |
| Q1+Q4 | 0.65 | 0.64 | 0.81 | 0.64 |
| Q2+Q3 | 0.35 | 0.36 | 0.19 | 0.36 |

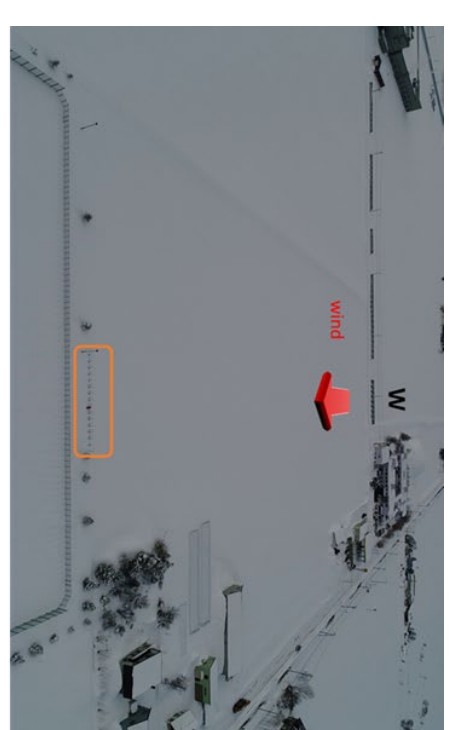

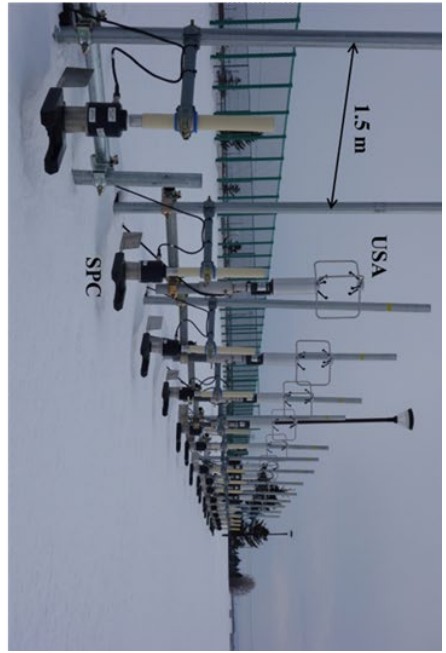

Figure 1. Observation field: Tobetsu, Hokkaido, Japan. the measurement zone is indicated with an orange frame (left). Measurement apparatus: fifteen Snow Particle Counters (SPCs) and Ultra Sonic Anemometers (USAs) set on towers 1.5 m apart (right).


**Figure 2. Meteorological conditions at the observation site in February 2018. Red arrows indicate the duration of the intense observation period.**



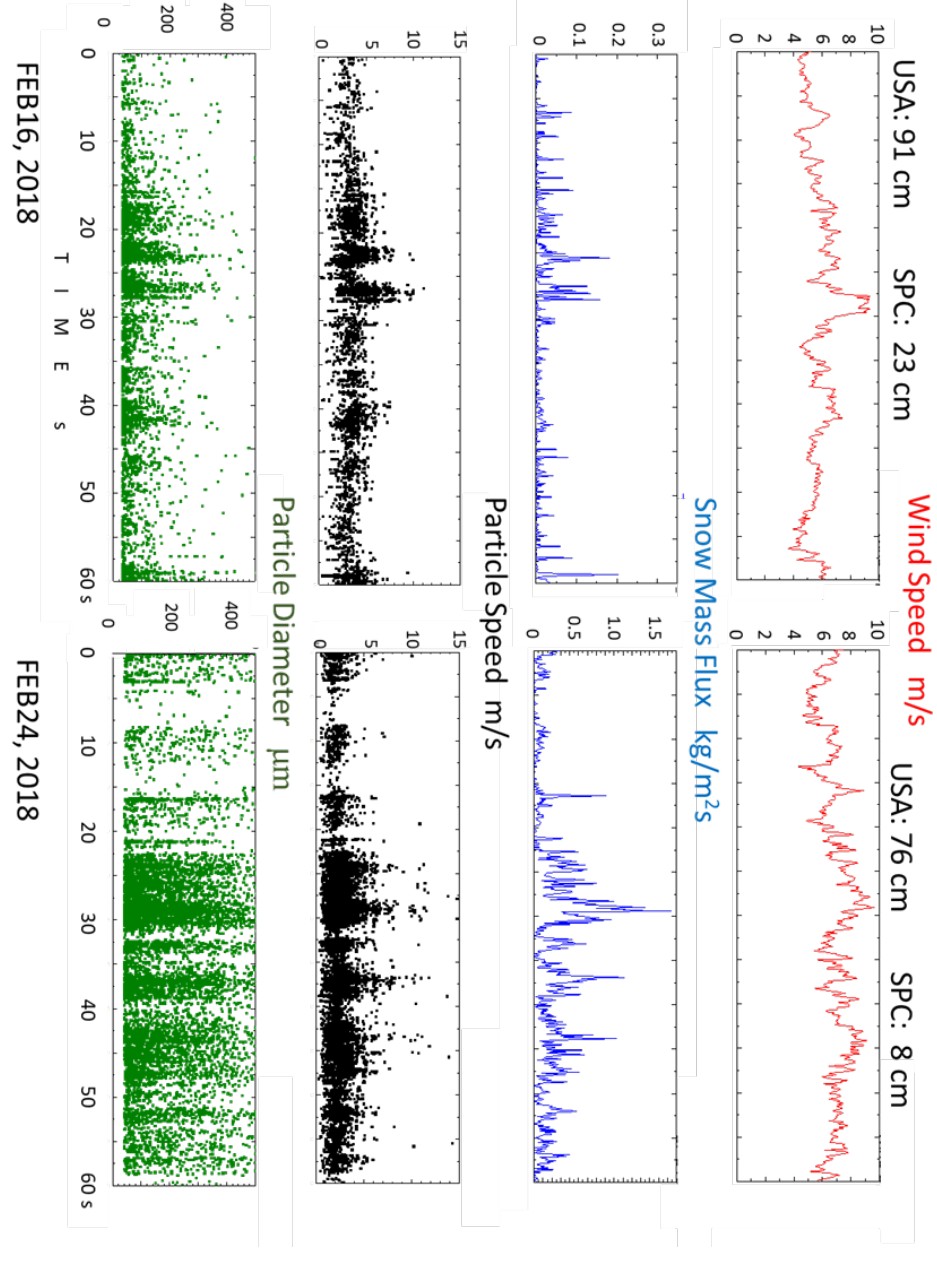

**Figure 3. Time series of wind speed, snow mass flux, particle speed, and particle diameter at the spanwise direction Y = 9 m. a: on February 16, the sensor heights for the USA and SPC were 91 cm and 23 cm, b: on February 24, 76 cm and 8 cm.**

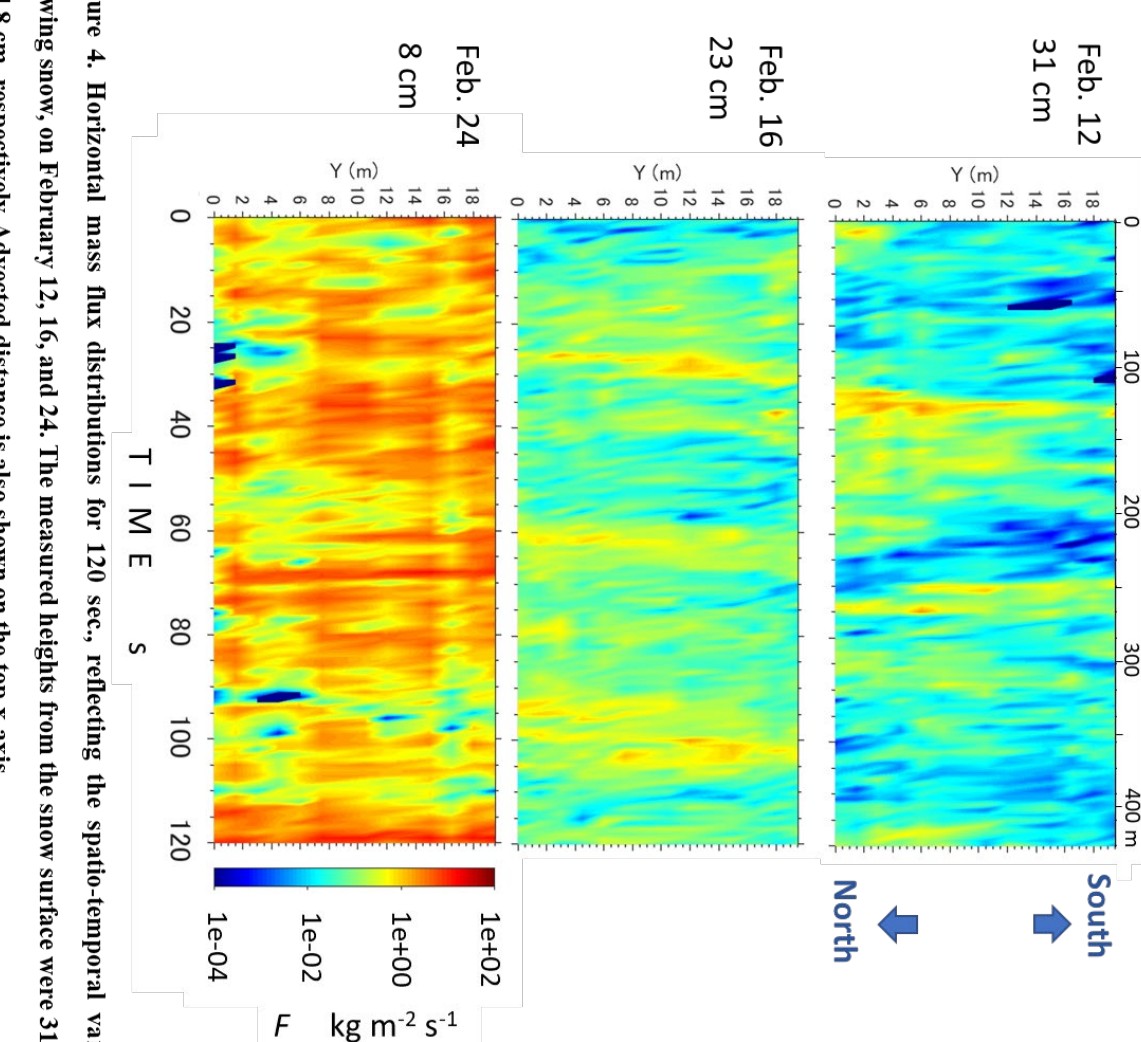

Figure 4. Horizontal mass flux distributions for 120 sec., reflecting the spatio-temporal variability of blowing snow, on February 12, 16, and 24. The measured heights from the snow surface were 31 cm, 23 cm, and 8 cm, respectively. Advected distance is also shown on the top x-axis.

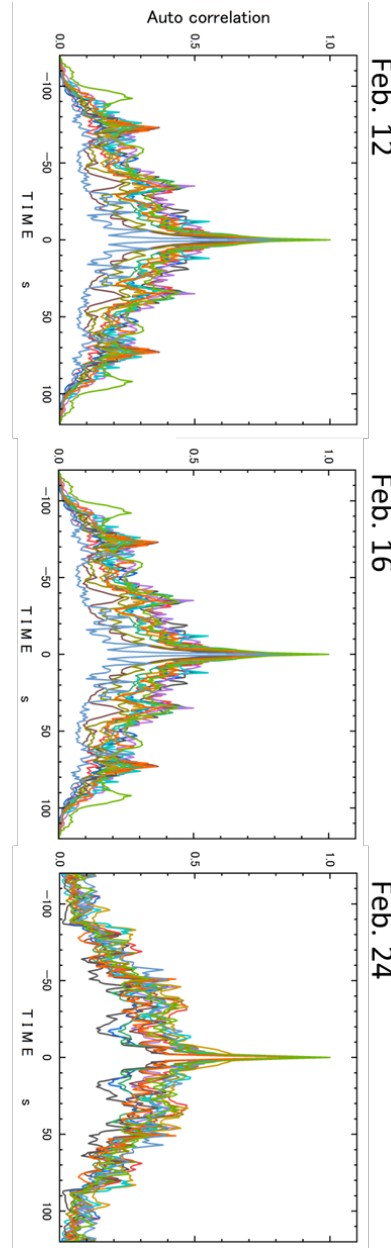

Figure 5. Autocorrelations of horizontal mass flux shown in Fig. 4.
Each line indicates the output from the individual SPC.



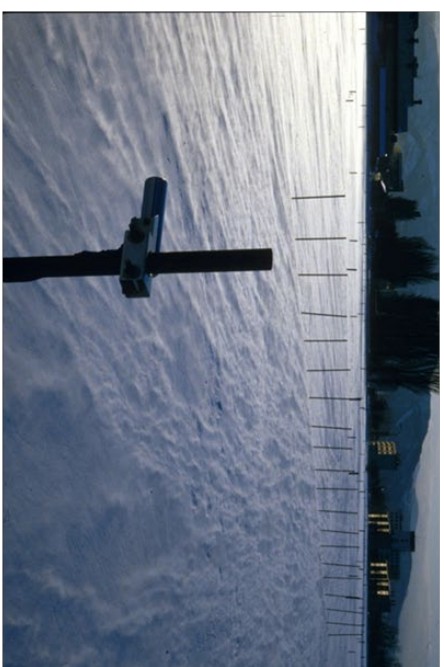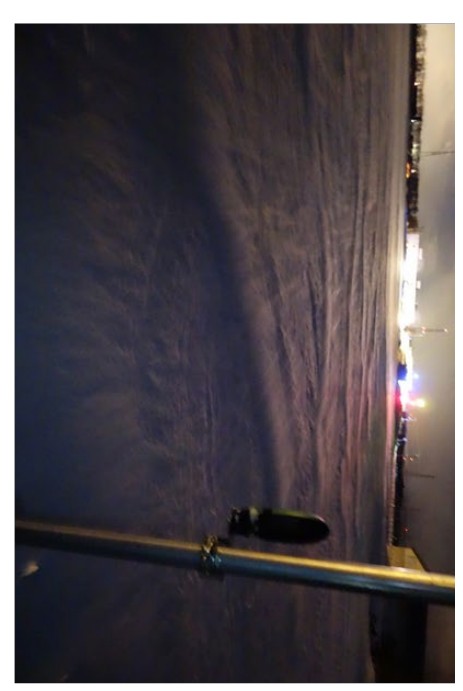

**Figure 6. 'Snow waves' appeared on the field's upstream at night on February 20 (above) and 'snow snake' (below). In both cases the wind blew from back to front. As shown in figures, the snow waves are organized in a lateral or spanwise orientation, while the snow streamers are quasi-parallel to the streamwise direction.**


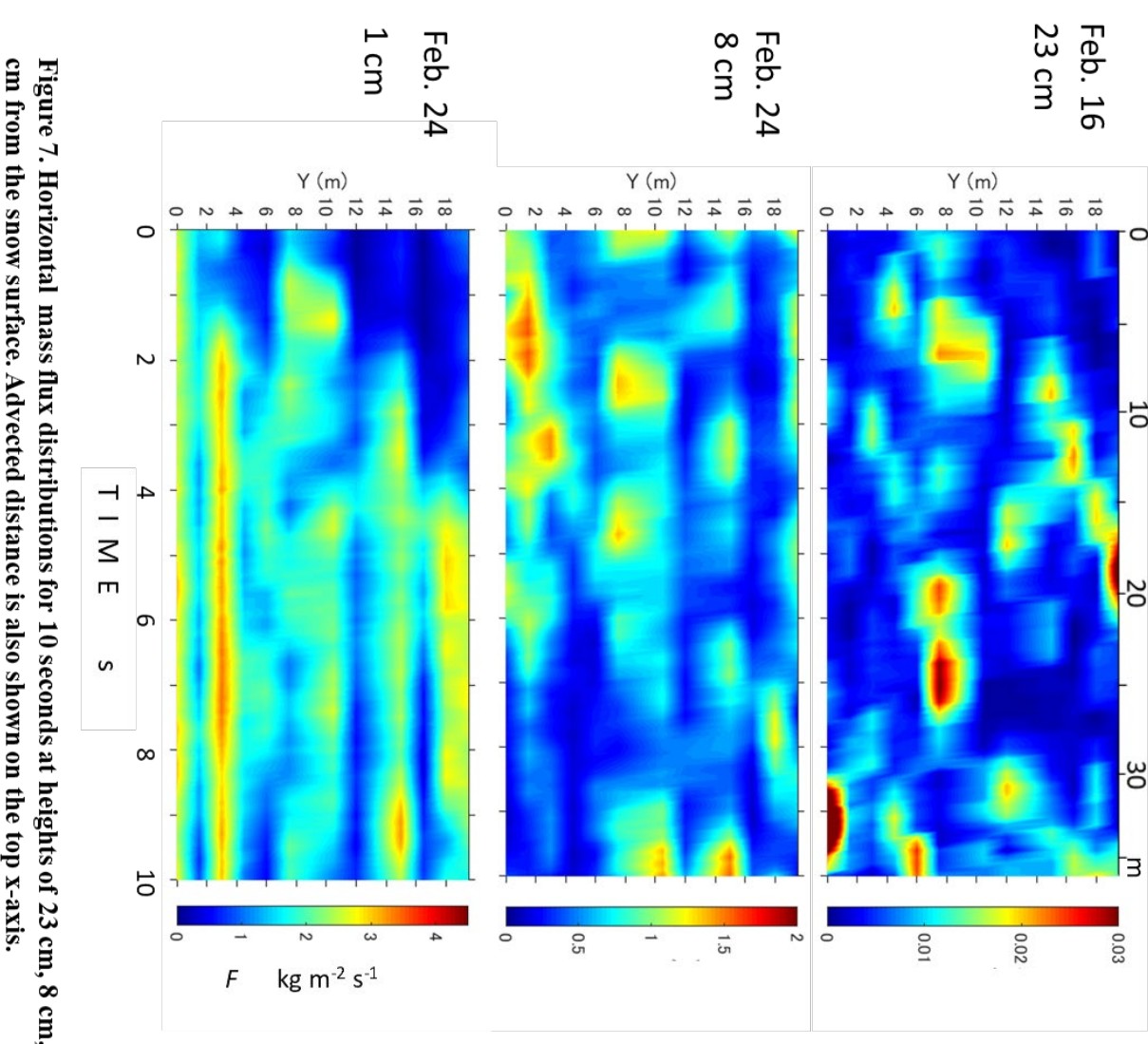

**Figure 7. Horizontal mass flux distributions for 10 seconds at heights of 23 cm, 8 cm, and 1 cm from the snow surface. Advected distance is also shown on the top x-axis.**


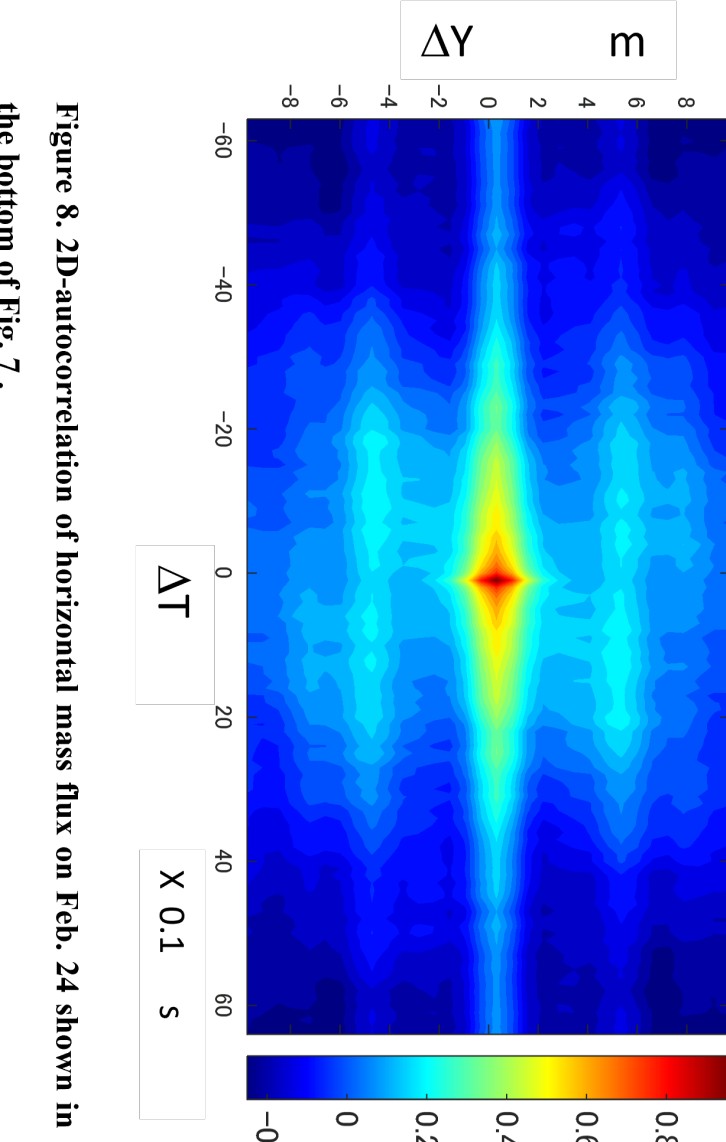

**Figure 8. 2D-autocorrelation of horizontal mass flux on Feb. 24 shown in the bottom of Fig. 7 .**


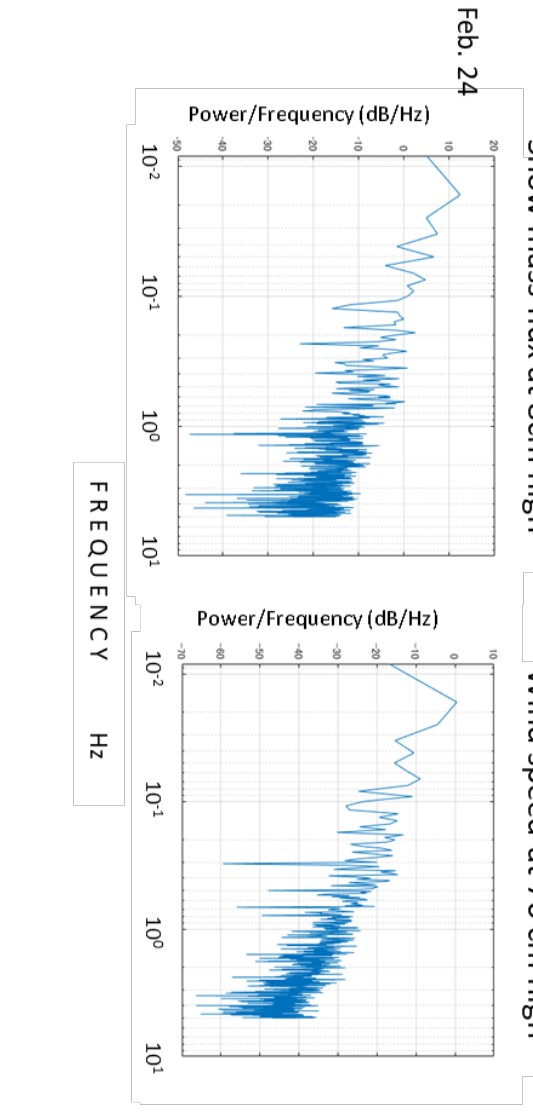

**Figure 9. Power spectra of mass flux and wind speeds at Y=10.5 m on February 24**


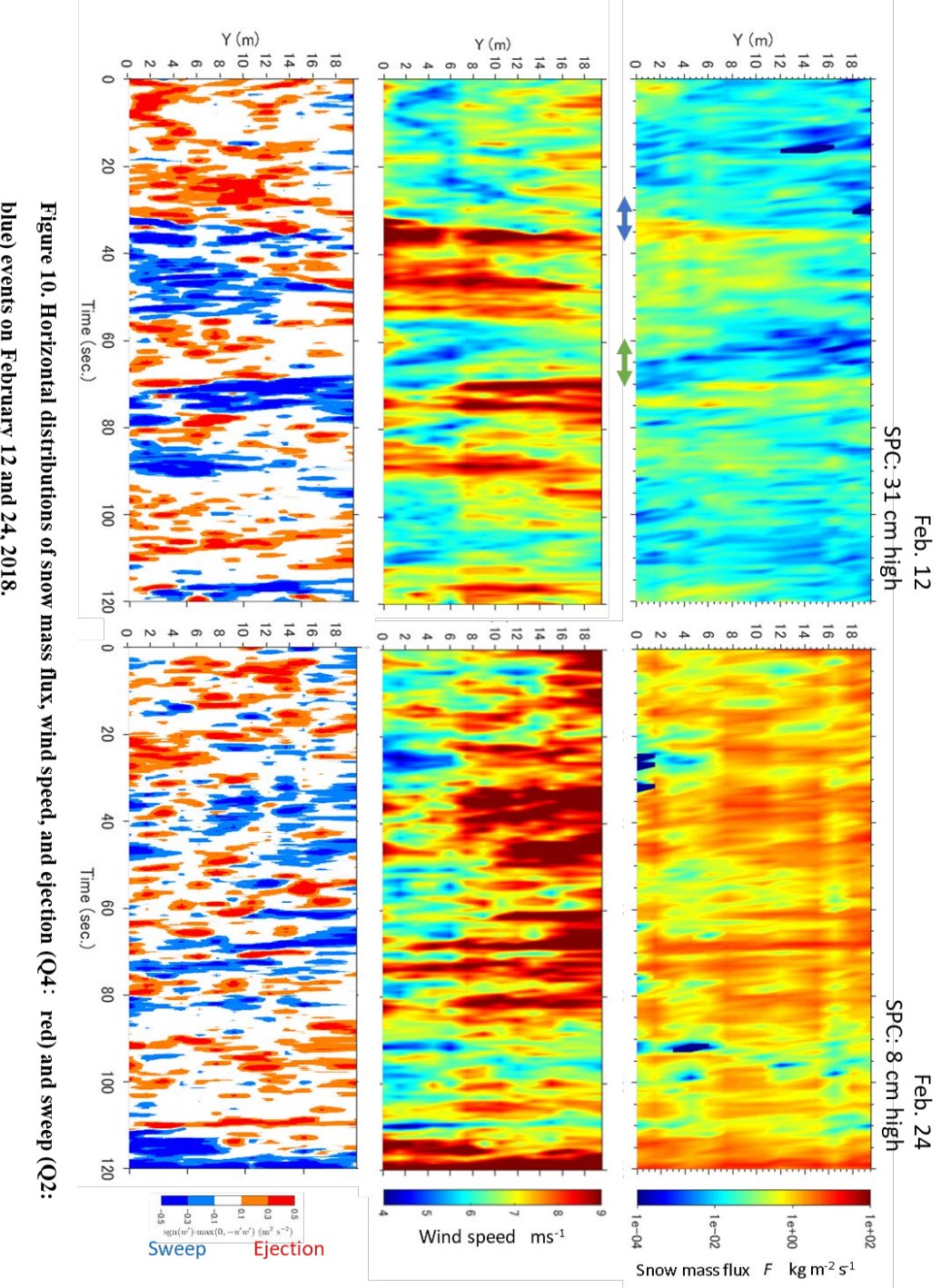

Figure 10. Horizontal distributions of snow mass flux, wind speed, and ejection (Q4: red) and sweep (Q2: blue) events on February 12 and 24, 2018.


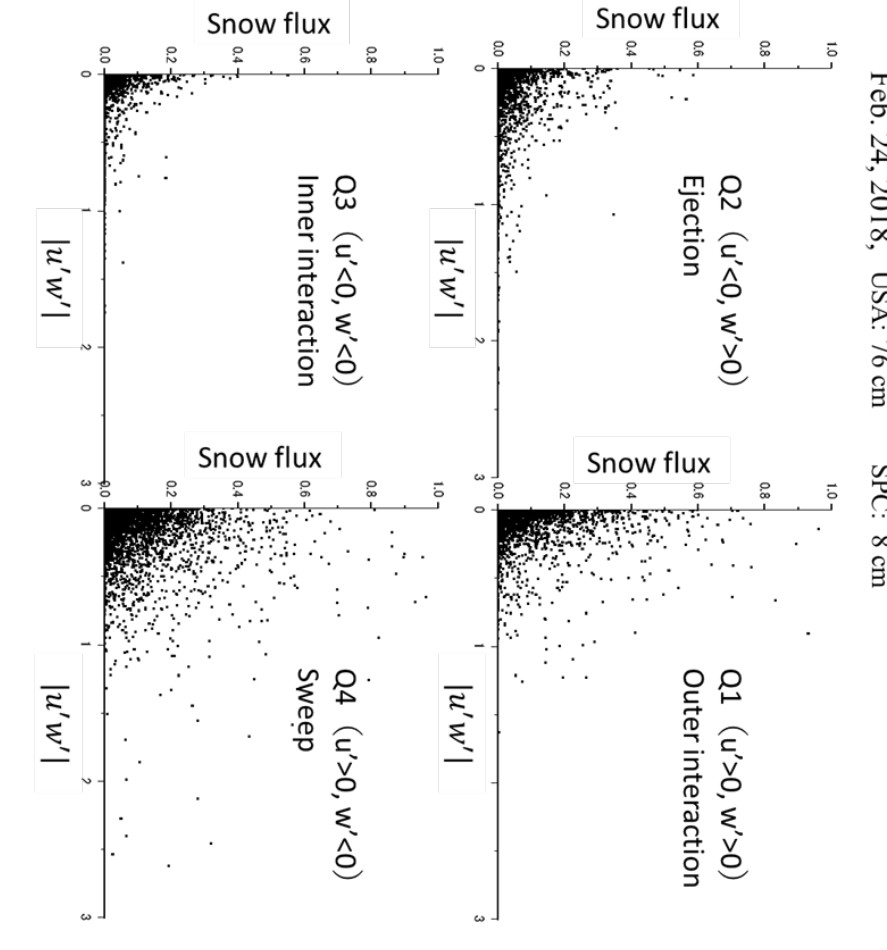

**Figure 11. Correlation between the horizontal snow flux at 8 cm and the absolute value of the kinetic shear stress *u'w'* in each quadrant on Feb. 24, 2018.**


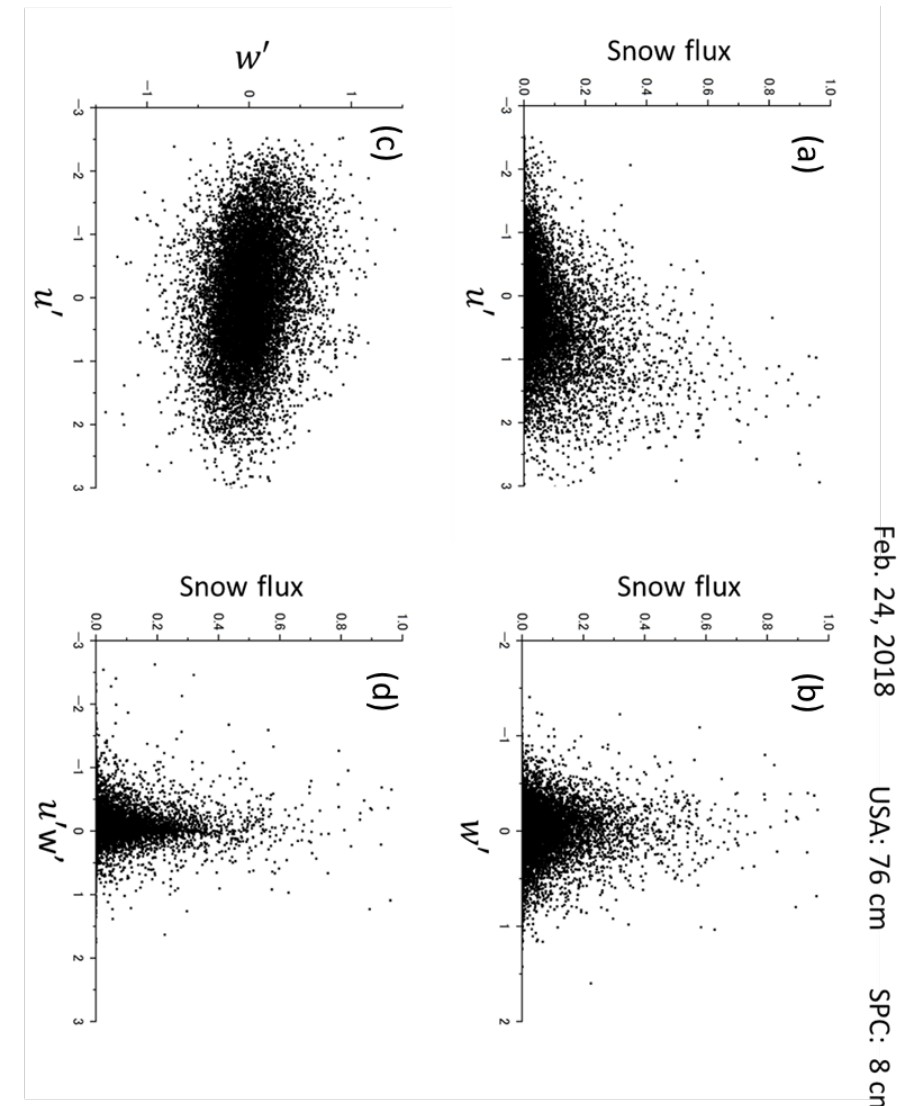

**Figure 12. Relations between the snow flux, the fluctuating component of *u'* and *w'*, and the product of *u'* and *w'* on Feb. 24, 2018. All the data and the observation period are the same as Figure 11.**


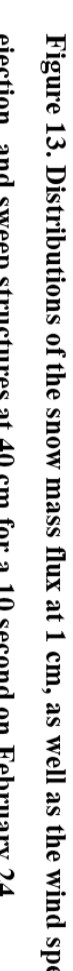

**Figure 13. Distributions of the snow mass flux at 1 cm, as well as the wind speed, ejection, and sweep structures at 40 cm for a 10 second on February 24.**

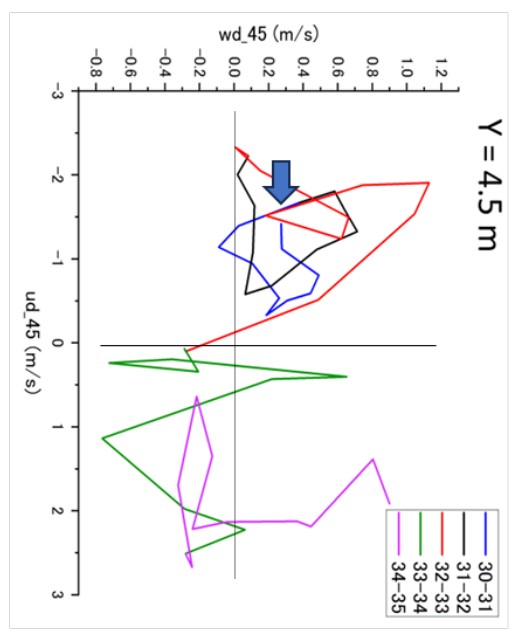
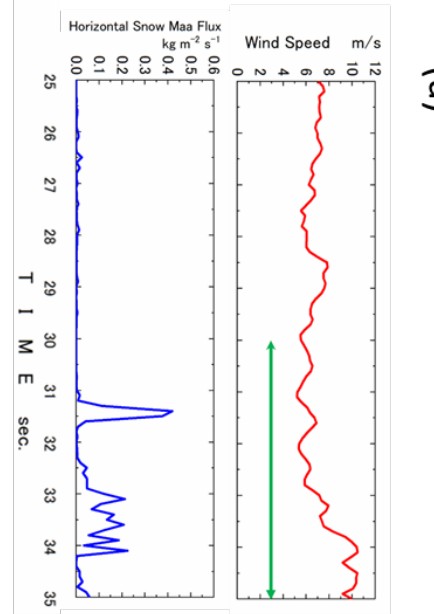

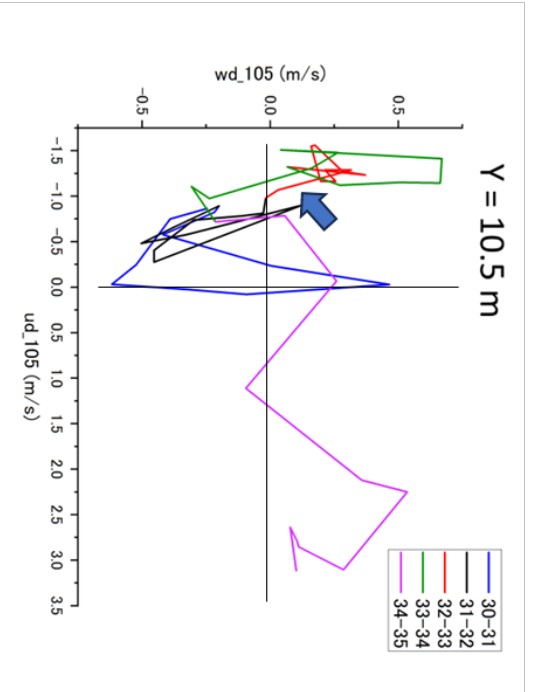
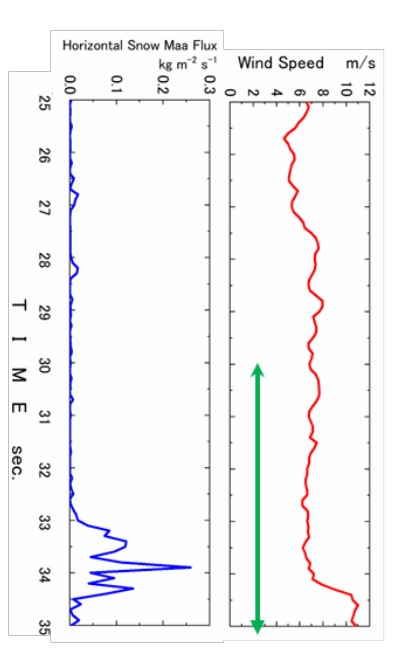

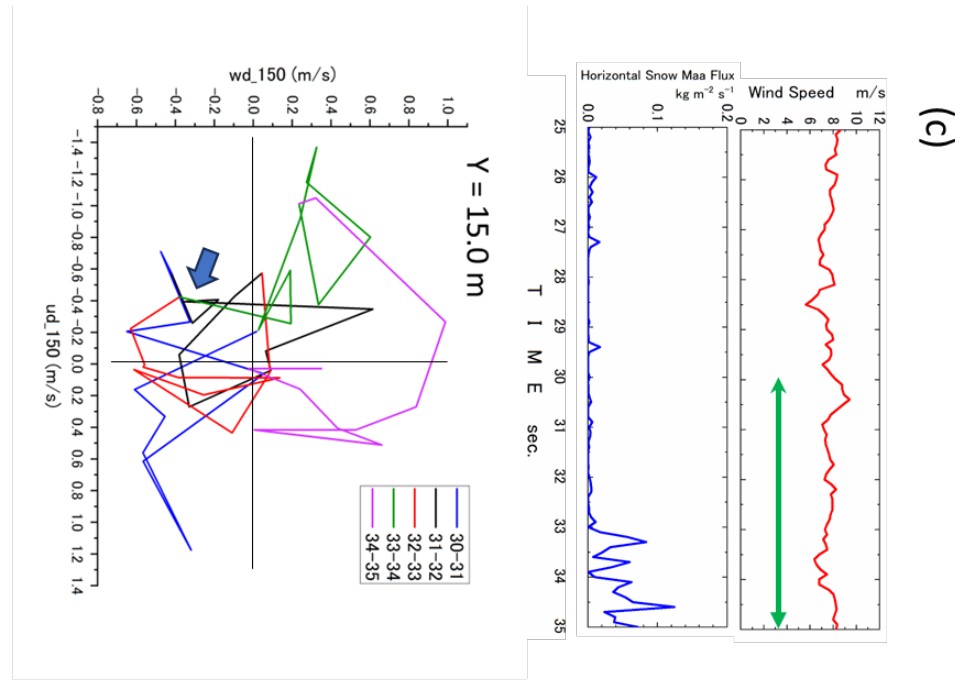

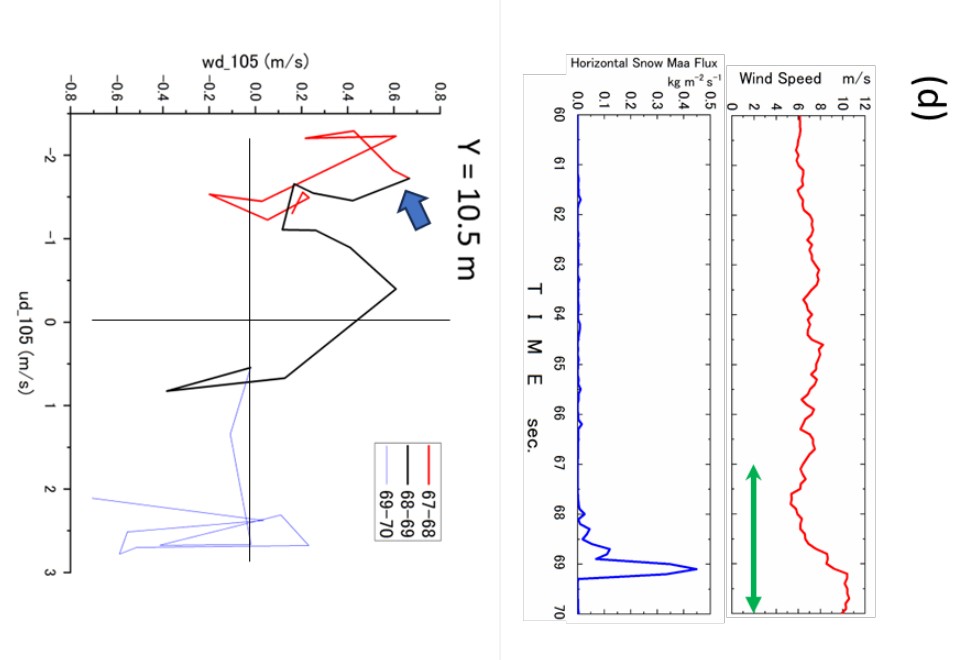

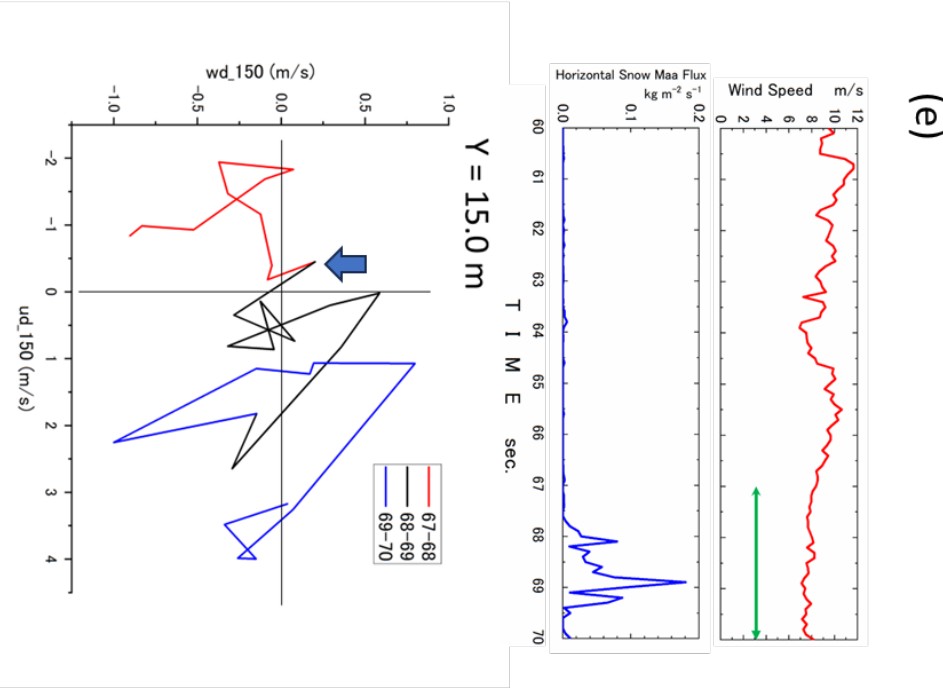

Figure 14. (a) to (e) present the wind speeds and mass fluxes at Y=4.5 m, 10.5 m, and 15.0 m from 25 to 35 s and at Y=10.5 m and 15.0 m from 60 to 70 s in Fig. 8. Furthermore, parametric curves of (*u'*(t), *w'*(t)) were plotted every one second with different colours for the period indicated by arrows in each figure: from 30 to 35 s for the former and from 67 to 70 s for the latter. The blue arrows indicate the onset of the snow flux increase.

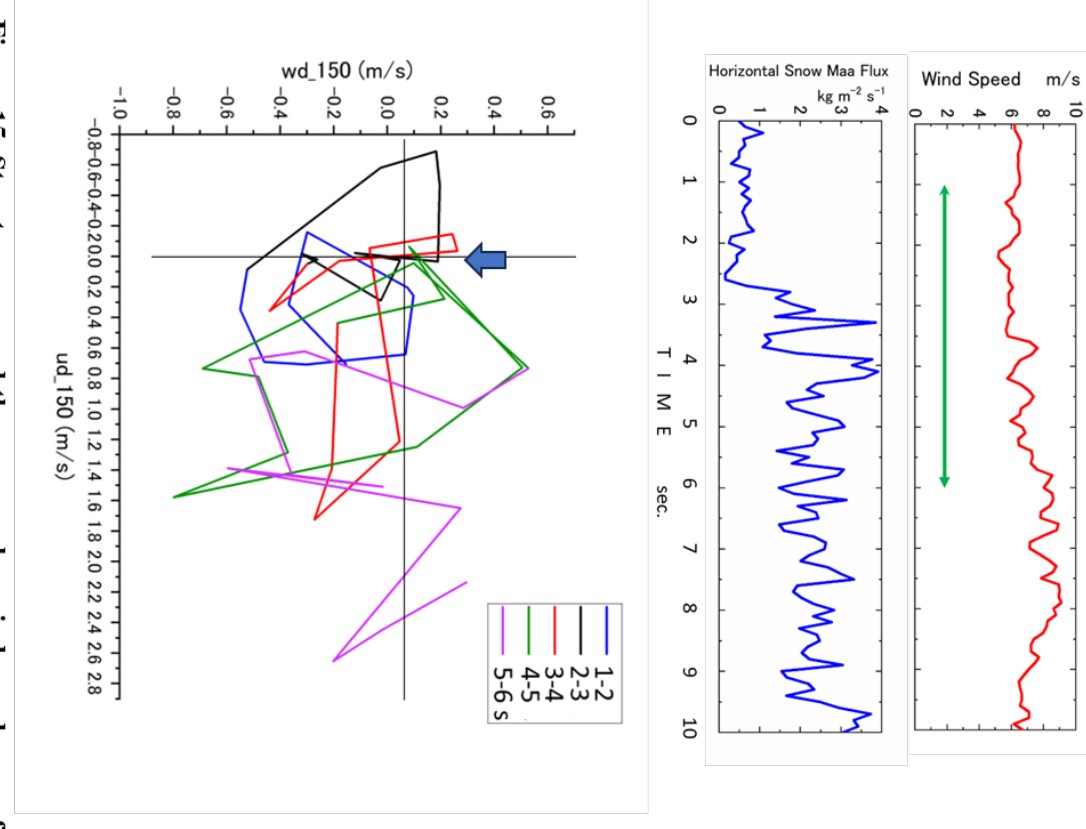

**Figure 15. Structures around the snow snake: wind speed, snow flux for ten seconds at Y=15.0 m, and the parametric curve from one to 5 s. The blue arrow indicates the onset of the snow flux increase.**