# Peer review of "Elucidation of Spatiotemporal structures from high-resolution blowing snow observations"

_EGUsphere, 2023_

## Author Response (AR1)

**Dear Editor,**

I sincerely appreciate your meticulous attention to our manuscripts. As is expressed in the previous rely, your dedicated efforts, particularly in the prolonged process of reviewer identification spanning almost three months, are genuinely valued.

As pointed out by the editor, it is crucial to acknowledge the fundamental distinctions between snow and sand particles. Notably, the particle density of sand surpasses that of snow by more than two-fold, significantly influencing parameters such as the wind threshold for particle transport initiation and the relationship between transport mass flux and wind speed. Additionally, snow, characterized by its cohesive nature, requires consideration of its impact at higher temperatures near zero when deliberating on the threshold wind speed for blowing snow initiation and development. However, given that the meteorological conditions outlined herein hover around -10 degrees Celsius and the phenomenon under scrutiny is already in the domain of 'snow in motion,' it is reasonable to disregard the sintering effect in this context. I firmly assert that blowing snow and sand transport fundamentally represent analogous facets of aeolian particle transport.

I extend my sincere appreciation for the insightful comments and recommendations provided by the editor and the two referees. Each contribution has been not only enlightening but also enriching. In response to invaluable suggestions, we have revised the manuscript as much as we could. It's worth noting that this manuscript primarily serves as a concise introduction to our distinctive observations and the resultant findings. While we endeavor to implement your suggestions comprehensively, I hope you understand that the sheer volume of data obtained from our extensive observations presents a multitude of avenues for a more in-depth exploration of the spatiotemporal structures of blowing snow. Specific aspects, such as the particle speed and wind speed in the proximity of these structures, as astutely pointed out by the editor, are topics we are actively addressing in subsequent manuscripts that are currently in progress.

Best regards,

Kouichi

**Dear Reviewer1,**

Thank you once again for your meticulous review of our manuscript and for your positive evaluations. As I mentioned in our previous correspondence, it is important to note that our primary objective with this manuscript is to provide a succinct overview of our distinctive observations and the results obtained. It is worth highlighting that the dataset resulting from our extensive observations is substantial, offering numerous opportunities to draw definitive conclusions that illuminate the spatiotemporal structures of blowing snow. More comprehensive analyses will be undertaken in subsequent manuscripts, which are currently in progress.

We have included the following description in the manuscript:

L47-48:

While the dataset obtained from our series of observations is indeed substantial, this study primarily offers a brief overview of these observations and the results obtained.

L246-250:

The dataset obtained from the series of observations in this study is substantial, providing ample opportunities to derive concrete conclusions elucidating the spatiotemporal structures of blowing snow. However, it's important to note that our analysis is still quite limited at this stage, and only preliminary results introduced here. Nevertheless, more detailed analyses, which will be presented in subsequent manuscripts, hold the promise of significantly contributing to the improvement and validation of the model.

We are deeply grateful for all of your insightful comments and suggestions, which have been immensely informative and educational. In accordance with your recommendations, we have undertaken significant revisions to the manuscript in collaboration with our coauthors.

Specifically, we conducted autocorrelations to determine the average streamwise and spanwise spacing of the observed features, as indicated in Figs 5 and 8. Subsequently, we have included the following descriptions in the manuscript.

L114-117:

The autocorrelations of horizontal mass flux in Fig. 4 are depicted in Fig. 5. Each line in the figures represents the output from an individual SPC. While the distributions on Feb. 20 appear somewhat subdued, distinctive peaks are evident on Feb. 12 and Feb. 16, occurring approximately every 10 s, 35 s, 70 s, and 90 s. These correspond to distances of approximately 30 m, 115 m, 230 m, and 300 m, respectively.

L130-134:

Figure 8 illustrates the 2D-autocorrelation of horizontal mass flux on Feb. 24, depicted at the bottom of Fig. 7. Despite the 1.5-meter spacing between SPCs, which did not offer sufficient resolution for clarity in the figure, the analysis suggests that the structures at 1 cm above the surface had widths on the order of 2 meters, with the peak of flux around 30 cm wide and a lateral spacing of about 5 meters. The strong correlations are evident within plus/minus $\Delta Y = 5$ m in Fig. 8.

The reason why we applied the quadrant analysis in this analysis and set on turbulent sweeps and ejections were additionally explained as follows.

L141-145:

We applied quadrant analysis in this study and focused on turbulent sweeps and ejections to better understand the spatiotemporal structures of blowing snow. While previous discussions have centered on the amount of snow transport by wind, often using averaged wind speed over specific time periods, such as the cube to 5th of mean wind speed for ten minutes (e.g., Dynin, 1967; Mann et al., 2000; Nishimura and Hunt, 2000). However, in order to elucidate the spatiotemporal structures of blowing snow, it is indispensable to set on our focus on the turbulent structures of the wind.

The formation mechanism of snow waves, as discussed by Kobayashi (1980), has been included in the manuscript. It is evident that both the formation mechanism of snow waves and snow snakes are closely tied to the turbulent structure of the wind, as introduced in this manuscript. However, at this stage, I do not have a specific idea to explain the formation mechanism of both phenomena or how we can distinguish between them. Following explanation was added in the manuscript.

L120-123:

Although the formation mechanism of the snow wave has not been clarified yet, Kobayashi (1980) suggested that snow waves strongly reflect the turbulent flow structure of the wind, and their formation mechanism is akin to the wavelike cloud formations in the atmosphere and the wind waves observed in the ocean.

**Specific Comments:**

L26: I think you mean modeling of, not study of

Following your suggestion, we changed the description from 'study of' to 'modeling of'.

L32-33: Can you provide an explanation of why you think turbulent sweeps and ejections, specifically, are crucial to discuss? Why not a different kind of coherent turbulent feature? Can we not generalize

to say that we just need to take a structure-based view of snow transport given its intermittent dynamics and non-linear dependence on wind speed?

Yes, as pointed out by the referee, turbulent sweeps and ejections serve as examples. To avoid any misunderstanding, we have included additional clarification in the manuscript to emphasize their representation amidst the intermittent dynamics and non-linear dependence on wind speeds.

L33:

, and intermittent dynamics and non-linear dependence on wind speeds,

L36-37: Can you provide a citation here?
We added a publication below as the reference.
L37:
(Kajiya et al., 2001)

References
Kajiya, Y., Kaneda, Y., and Tanji, K.: Factors Inducing Multivehicular Collisions During Visibility Reduced by Snowstorm, Transportation Work-Zone Safety and Winter Services, Vol. 1745, Issue 1, 61-66. https://doi.org/10.3141/1745-08, 2001.

L46: Conducting is a different tense.
We have corrected the tense as 'conducted'.

L72: You have two sections titled "Results and analysis"
Sorry for the careless mistake. We changed the second one to 'Conclusions'.

L73: Are these average or instantaneous values you are plotting?
Description of 'ten minutes average of' is added in L88 of the manuscript.

L75: Westerly?
L90:
Sorry, we corrected the spelling error.

L79-80: Are you referring to conditions at the AWS? Please specify.
L94: The description of 'at the AWS site" was inserted in the sentence.

L88: Can you show us that the particles are generally slower than the wind?

As depicted in Figure 3, the velocities of most particles are observed to be lower than the wind speed recorded by the ultra-sonic anemometer (USA). Although the sensor height of the Snow Particle Counter (SPC) is lower than that of the USA, rough estimates based on the logarithmic wind profile support this trend.

Following explanation was added in the manuscript.

L104-105

Although the sensor height of SPC was lower than that of USA, rough estimates based on the logarithmic wind profile support this trend.

L91: Can you comment on what the range of diameters that is measurable by the SPC? Is this the upper end of that range?

Explanations about the SPC are largely expanded including the measurable particle size following the suggestion by another reviewer. Following expression was added in the manuscript.

L 58-66:

The SPCs used in this study (Niigata Denki Co.) is an optical device (Nishimura and Nemoto, 2005) that measures the diameter and the number of drifting snow particles by detecting their shadows on a photodiode (assuming that drifting particles are spherical in shape). Electric pulse signals resulting from snow particles passing through the sampling volume ($2 \times 25 \times 0.5$ mm) are sent to a transducer and an analyzing data logging system (PC). In this way, the SPC is able to detect particles in the range of 40–500 µm. The analysis software divides the particles into 64 size classes and records the number of particles in each size class at 1-s intervals. The SPC is mounted on a self-steering wind vane, and hence the sampling region, which has a cross-sectional area of $2 \times 25$ mm ($50$ mm$^2$), is maintained perpendicular to the horizontal wind vector. If the diameter of a snow particle is larger than that of the maximum diameter class, the snow particle is considered to belong to the maximum diameter class. Usually, SPCs are used to obtain snow particle size distribution and mass flux at 1-second intervals.

L106:

 '50 – 450 µm' to '40 – 500 µm'

L93: It would be very beneficial if in addition to time, you included an average "advected distance" on your x-axis. This could be something like you have mentioned with frozen turbulence: time x avg wind speed. That would really help us see the dimensions of the features you are describing. For example, is Figure 4 approximately 500 meters in the x-direction?

Following the suggestion, we put the "advected distance" on x-axis utilizing the average wind speed. Since the sensor height of SPC is lower than that of USA, assuming the logarithmic wind profile, the wind speed was estimated. The following description has been added to the manuscript.

L114 – L116:

In the figure, the 'advected distance' is presented on the x-axis, estimated using the average wind speed provided by the ultra-sonic anemometer (USA) and corrected for the height differential between the USA and the Snow Particle Counter (SPC) sensor heights, assuming the logarithmic wind profile.

L96: "Precipitation observed with DFIR was"

Thanks, we have corrected the grammatical error. (L111)

L97: Can you please rephrase the sentence beginning with "They show the maps…"

The following description has been added to the manuscript to explain the methodology used to generate the maps of snow transport intensity in Figure 4:

L111-114:

The three panels in Figure 4 depict maps of snow transport intensity, created by aligning time series data from the Snow Particle Counter (SPC) transverse array in the spanwise Y direction and applying Taylor's frozen hypothesis (1938) to substitute time for the streamwise direction.

L99: Westward wind direction or westerly wind?

L117:

Westerly wind is correct. We have changed the word accordingly.

L100: What was the period of the periodic changes?

Following description was put on L118.

'periodic time, in other words, lateral to the streamwise direction, variation'

L103: I believe your reference to Figure 5 is out of order. However, it would be beneficial to include a comparison with the photo, including an indicator of lengths.

The order is correct. The picture above shows the snow wave, and the one below is the snow snake. We put the additional words in the manuscript and figure caption.

L125: from back to front

Figure caption in Fig. 6: In both cases the wind blew from back to front.

L101-102: Here or later, can you please comment on what you think the snow waves are caused by?

I suppose that the formation mechanism of the snow wave has not been clarified yet. The sole explanation I know of is Kobayashi (1980).

Based on the USA measurements, he found that the wavelength of the snow waves was nearly the same order of magnitude as the scale of wind turbulence. Then, he suggested that the snow waves

strongly reflect the turbulent flow structure of the wind, and its formation mechanism is similar to the wavelike cloud in the atmosphere and the wind wave in the ocean.

The following description has been added to the manuscript at L126:
"Although the formation mechanism of the snow wave has not been clarified yet, Kobayashi (1980) suggested that the snow waves strongly reflect the turbulent flow structure of the wind, and its formation mechanism is similar to the wavelike cloud in the atmosphere and the wind wave in the ocean."

L112-115: This is a bit surprising and somewhat counter-intuitive. In the atmospheric boundary layer, we often assume that the size of the eddies governing the flow increase as we move away from the surface, but your presentation here suggests the opposite. Can you please comment on what you think is happening and why this is different from the conventional view?

Exactly. It is generally known that the size of the eddies increases with the height from the surface. Contradiction appeared in Figure 9 is probably because the date calculated the power spectrum are different. The specific point we would like to emphasize here is that the dominant frequencies and general trends of the power spectra for snow flux and wind speed appear very similar. So, we delete both the pictures of Feb. 12 and Feb. 16, and the following description has been revised and added to the manuscript:

L142-145:

"Figure 9 displays the power spectra of mass flux at Y=10.5 m for February 24 in Fig. 4, along with the ones for wind speeds. The dominant frequencies and general trends of the power spectra for snow flux and wind speed appear very similar, implying that both vary in a correlated manner. Despite the sensor heights of SPCs being 31 cm lower than those of anemometers, these observations suggest that changes in snow flux reflect the structures of turbulence eddies near the snow surface."

L116: Can you report what the dominant wind frequency is versus the dominant snow frequency? As well, at the higher end of the frequency spectrum, did you see any influence of stochastic particle-surface interactions that weren't present in the wind?

While the existence of snow particles in the flow may indeed affect the structure of turbulence, further investigation is required to fully understand this phenomenon. Bintanja (2000) suggested a damping of turbulence in the presence of suspended snow particles. However, due to the complexity of the interaction between snow particles and turbulence, a detailed analysis is needed to explore this issue thoroughly.

L119-120: Can you provide a reference for quadrant analysis, such as Wallace's review from 2016.

Following the suggestion, we added Wallance (2016) in L151 and the reference.

Wallance, J. M.: Quadrant Analysis in Turbulence Research: History and Evolution, Annual Review of Fluid Mechanics, Vol. 48, 131-158 , https://doi.org/10.1146/annurev-fluid-122414-034550, 2016.

L125: What quadrant hole size did you use?

In this study, we did not specify the hole size in Figures 10, 11, and 13; it was assumed to be zero. Practically, we can consider the size to be 0.1 in Figs 10 and 13, as indicated by the color bar ranging from -1 to 1. While it would be interesting to investigate how the contributions of ejection and sweep change with different hole sizes, we leave this issue for future consideration.

Figure 5: It would be helpful to note for those unfamiliar that the snow waves are organized in a lateral or spanwise orientation and the snow streamers are quasi-parallel to the streamwise direction.

Thank you. Following the suggestion, explanations about the snow waves and the snow streamers are additionally provided in the caption of Fig. 6, as shown below.

In both cases the wind blew from back to front. As shown in figures, the snow waves are organized in a lateral or spanwise orientation, while the snow streamers are quasi-parallel to the streamwise direction.

Figure 7: Can you include the heights of the measurements in the labels.

We showed the measurements height of snow flux and wind speed in Fig. 9.

Figure 8: This is a very nice figure and data that has never been available before. You appear to be using multiple fonts. As well, it could improve the presentation to write "Sweep or Ejection" to maintain symmetry with the adjacent colorbars.

We appreciate very much. We unified all the characters in the figure and changed the position of "Sweep or Ejection" as well.

L127: We can't actually see that the snow fluxes were predominantly observed in Quadrant 1 in the figure because we have no idea the density of the dots.

Sorry, it is probably hard to recognize in the figure. Could you look at Table 1 in which all the contributions are indicated.

Figure 9: There are some curious features here. For example, this looks like an 1/x relationship between Reynolds stress magnitude and snow flux for all quadrants. Do you have any comment on why the highest snow fluxes appear to be when the magnitude of u'w' is on the lower end, and why snow fluxes taper off with larger u'w'? For those that use u'w' to calculate friction velocity (u*) in

blowing snow models, this should be a very disturbing finding.

Given your comments about Q1 and Q4 being dominant, can you create a plot comparable to Figure 9 that show the relationship between windspeed magnitude and snow flux? Is there still a 1/x scaling? If there is a stronger correlation, perhaps a "gust-factor" type approach would make sense to revisit (rather than u* or u'w') for future blowing snow model development?

Figure 11 suggests a decrease in snow flux with increasing Reynolds stress, which may initially appear contradictory to conventional understanding. To further explore this relationship, Figure 12 examines the correlation between snow flux and the fluctuating components of u' and w'. Subfigure (a) demonstrates a clear increase in snow flux with u', while Subfigure (b) illustrates a distribution concentrated around zero, resembling a normal distribution. Subfigure (c) reveals a negative trend between u' and w', indicating that w' decreases as u' increases. The relationship between the product of u' and w' and snow flux is depicted in Subfigure (d). Although setting specific hole sizes in Figure 11 may result in variations, Leenders et al. (2005) suggested that vertical fluctuations are inherently constrained by the distance from the bottom boundary and the overall scale of the structure, leading to poor correlation between surface fluctuations and vertical fluid motions higher in the profile. Consequently, measurements of Reynolds shear stress around 1 m from the surface are typically poorly correlated to snow transport flux.

Above discussions are inserted in the manuscript from L166-175.

L130: Why are you specifying these features as "Ejection-like" and "sweep-like" here?

We have changed the description as follows.

L162:

"Ejection" and "sweep"

Thank you very much!

L180-189: Your results and Bauer's results appear to be in contradiction as they reported Q2 and Q4 dominance, but you report Q1 and Q4, but you state that they align. Can you help me understand?

We have changed the description in the manuscript as follows.

L221-231

Sterk et al. (1998) utilized a Gill-type anemometer and acoustic sediment sensing instruments to investigate the relationship between sediment flux and Reynolds shear stress. They reported very low levels of statistical explanation, with correlation coefficients smaller than 0.14, which aligns with our findings, as depicted in Fig. 12(d). Furthermore, Sterk et al. (1998) partitioned instantaneous sediment transport values and sorted them according to wind speed quadrant signatures. They observed that the

largest mean saltation fluxes occurred during stress excursions located in Q1 and Q4 quadrants, while excursions in the Q2 and Q3 quadrants were unable to sustain appreciable saltation activity. This observation is consistent with the findings of Shonfeldt and von Lowis (2003), Leenders et al. (2005), and Wiggs and Weaver (2012) who employed sonic anemometers for sand transport observations. Our study corroborates these findings, as summarized in Table 1, where contributions from Q1 and Q4 quadrants were significantly larger than those from Q2 and Q3. This supports the notion that horizontal velocity fluctuations (u') play a pivotal role in aeolian sediment transport compared to vertical fluctuations (w') or kinematic shear stress (u'w').

L183: I believe these studies were on sand, and not snow transport.

Yes, Bauer et al. (1998) conducted the research on sand transport. I believe the blowing snow and sand transport are essentially the same phenomena as the aeolian particle transport. We provided additional words below in the manuscript to specify that Bauer et al. (1998) is done for the sand transport.

L218:

for the sand transport,

L204-206: Please rephrase this sentence. It appears incomplete.

Sorry, we have corrected the description as follows.

L246-252

Indeed, conducting detailed measurements with SPCs and USAs positioned at the same height from the surface would provide more accurate and directly comparable data. Additionally, incorporating the latest turbulence analysis techniques could further enhance our understanding and facilitate reaching conclusive findings. By leveraging advanced methodologies and instrumentation, such as high-resolution SPC arrays and state-of-the-art turbulence analysis algorithms, we can obtain more precise measurements and delve deeper into the complex interactions between blowing snow and turbulent flow dynamics. These efforts would contribute significantly to advancing our knowledge in this field and ultimately lead to more comprehensive conclusions.

Figure 10: Do you have any idea why the massflux appears constant at y=0 in your top panel? It appears in your sweep/ejection panel that the ejections are preceding the start of transport. Does this support the idea of a bottom-up mechanism driving snow transport initiation? That is, the idea that eddies pull away from the surface (causing an ejection) and an in-rush of air fills the space (causing a sweep)?

Thank you for providing this clarification. Setting the first SPC sensor at Y=0, positioned at the bottom of the figure, allowed for the detection of a certain amount of snow flux at that level. While the flux appeared relatively constant in the previous figure, adjusting the range of the color bar has now made

the temporal variation more apparent. This adjustment has provided a clearer understanding of how the snow flux changes over time at this particular sensor location.

Figure 11/12: It would be very helpful if you could mark the onset of snow transport in your parametric curves.

We put the arrow(s) in Figures 14 and 15, that indicate the onset of the snow flux increase. Further, following description was put additionally in both figure captions.

The blue arrow(s) indicate the onset of the snow flux increase.

**Dear Reviewer2,**

We deeply appreciate all of your insightful comments and suggestions, which have proven to be both informative and educational. In line with your recommendations, we have carried out substantial revisions to the manuscript in collaboration with our coauthors.

As we explained in the previous reply, please understand that, in this manuscript, our intention is to provide a brief overview of our unique observation and the obtained results. As pointed out by the reviewer, our analysis is still quite limited, and we are presenting preliminary results. More comprehensive analyses will be deferred to subsequent manuscripts, currently in progress."

We put the following description in the manuscript.

L47-48:

While the dataset obtained from the series of observations is substantial, this study aims to provide a concise overview of the observations and the results obtained."

L254-258:

The dataset obtained from the series of observations in this study is substantial, offering various approaches to derive concrete conclusions explaining the spatiotemporal structures of blowing snow. Although our analysis is still limited at this stage, and only preliminary results are introduced here, more detailed analyses, which will be presented in subsequent manuscripts, will significantly contribute to the improvement and validation of the model.

The Snow Particle Counter (SPC) serves as a well-established blowing snow sensor, providing outputs of both particle size and numbers, which represent transport flux every second. Interestingly, it has also found application in sand transport research as a "Sand Particle Counter." Moreover, prior to experimentation, thorough calibrations of all sensors are conducted to ensure accuracy, thereby reasonably mitigating the effects of sensitivity differences.

Following the suggestion, expanding on the experimental setup, particularly concerning the SPC, we have included detailed explanations. Notably, due to snow accumulation or settling, sensor heights may vary over time. To address this, we meticulously measured the height manually at each observation period, ensuring consistency and reliability in our measurements.

Following description was added on the manuscript.

L58-66:

The SPCs used in this study (Niigata Denki Co.) is an optical device (Nishimura and Nemoto, 2005) that measures the diameter and the number of drifting snow particles by detecting their shadows on a photodiode (assuming that drifting particles are spherical in shape). Electric pulse signals resulting

from snow particles passing through the sampling volume (2 × 25 × 0.5 mm) are sent to a transducer and an analyzing data logging system (PC). In this way, the SPC is able to detect particles in the range of 40–500 μm. The analysis software divides the particles into 64 size classes and records the number of particles in each size class at 1-s intervals. The SPC is mounted on a self-steering wind vane, and hence the sampling region, which has a cross-sectional area of 2 × 25 mm (50 mm2), is maintained perpendicular to the horizontal wind vector. If the diameter of a snow particle is larger than that of the maximum diameter class, the snow particle is considered to belong to the maximum diameter class. Usually, SPCs are used to obtain snow particle size distribution and mass flux at 1-second intervals.

L69-70:
Accurate calibrations of all sensors are carried out beforehand, allowing us to reasonably exclude the effect of sensitivity differences.

L72-76:
As elucidated further in subsequent sections, the heights of the sensors from the surface fluctuated due to snow accumulation. Consequently, we took the necessary step of manually measuring the sensor heights of both the Snow Particle Counters (SPCs) and Ultrasonic Anemometers (USAs) at every observation period. This meticulous approach ensured that any variations in sensor height were accurately accounted for, maintaining the integrity and reliability of our measurements throughout the experiment.

We utilized data from all sensors in generating Figures 4, 7, 10, and 13, which depict the temporal and spatial distributions of snow flux and wind. Additionally, Table 1, illustrating the contributions of each quadrant, was derived from data collected by all sensors. Given the significant increase in snow depth, approximately 10 cm within a single day on Feb. 24, as depicted in Fig. 2, we present two scenarios with SPC heights set at 8 cm and 1 cm. This variation in sensor height accounts for the changing snow accumulation, ensuring our analysis comprehensively captures the evolving environmental conditions.

We inserted the sentence below in the manuscript.
L130-131:
Due to the substantial increase in snow depth, nearly 10 cm within a single day on Feb. 24, as depicted in Fig. 2, we have introduced two scenarios with SPC heights set at 8 cm and 1 cm.

L162:
analysis was conducted with all the sensor data.

Indeed, we have gathered substantial data; however, for this manuscript, we opted to present only typical data. This decision partly explains why we depicted data for relatively short periods of time. Analyzing longer spans of data undoubtedly yields further valuable insights, but we have chosen to defer this to future investigations. As rightly noted by the reviewer, wind direction variability can significantly influence our observations. While the SPCs can maintain perpendicularity to the horizontal wind vector, we took care to select data points when the wind direction remained constant.

In practice, the 1.5-meter spacing in our measurements may not be fine enough to discern the precise structures described here, particularly the streamers; they appeared rather bold compared to our expectations. Additionally, in the previous figures, the flux appeared almost constant. However, by adjusting the range of the color bar in the figures, the width of the streamers became reasonably thinner, and the temporal variations became clearer in Figs. 7 and 12. Furthermore, based on a suggestion from another reviewer, we calculated the 2D-autocorrelation of horizontal mass flux and included the description as follows.

L135-139:
Figure 8 represents the 2D-autocorrelation of horizontal mass flux on Feb. 24, as depicted at the bottom of Fig. 7. Despite the 1.5-meter spacing between SPCs not providing sufficient resolution to be clearly visible in the figure, it suggests that the structures at 1 cm above the surface had widths on the order of 2 meters, with the peak of flux around 30 cm wide, and a lateral spacing of about 5 meters. The strong correlations are indicated by the plus/minus $\Delta Y = 5$ m range in Fig. 8.

Exactly. It is generally known that the size of the eddies increases with the height from the surface. The discrepancy observed in the previous Figure 7 may be due to differences in the data used to calculate the power spectrum. The specific point we would like to emphasize here is that despite these differences, the dominant frequencies and general trends of the power spectra for snow flux and wind speed appear very similar. Therefore, we have removed both the pictures of Feb. 12 and Feb. 16, as well as the part describing the change of the dominant eddies with height from the manuscript as follows.

L134-137:
Figure 9 displays the power spectra of mass flux at Y=10.5 m for February 24 in Fig. 4, along with the ones for wind speeds. The dominant frequencies and general trends of the power spectra for snow flux and wind speed appear very similar, implying that both vary in a correlated manner. Despite the sensor heights of SPCs being 31 cm lower than those of anemometers, these observations suggest that changes in snow flux reflect the structures of turbulence eddies near the snow surface.

Thank you very much for the valuable information. We included the work by Wiggs and Weaver (2012) in line 227 and reference.

---

## Editor Decision (ED1)

[revised manuscript text omitted]

(c)

[Figure]

(d)

[Figure]

**Figure 14.** (a) to (e) present the wind speeds and mass fluxes at Y=4.5 m, 10.5 m, and 15.0 m from 25 to 35 s and at Y=10.5 m and 15.0 m from 60 to 70 s in Fig. 8. Furthermore, parametric curves of $(u'(t), w'(t))$ were plotted every one second with different colours for the period indicated by arrows in each figure: from 30 to 35 s for the former and from 67 to 70 s for the latter. The blue arrows indicate the onset of the snow flux increase.

[Figure]

**Figure 15. Structures around the snow snake: wind speed, snow flux for ten seconds at Y=15.0 m, and the parametric curve from one to 5 s. The blue arrow indicates the onset of the snow flux increase.**

---

## Author Response (AR2)

**Dear Editor,**

I sincerely appreciate your kind and encouraging comments and suggestions regarding our manuscript. We have tried to revise the manuscript as thoroughly as possible.

For the comments from Reviewer 1, we corrected the typo as follows:

Lines 137 and 305: from "Dynin" to "Dyunin."

However, we are somewhat confused by the comments from the second reviewer. As I mentioned in my previous reply, this manuscript aims to present a concise overview of our unique observations and the obtained results. The dataset from our extensive observations is substantial, providing various avenues for concrete conclusions about the spatiotemporal structures of blowing snow. More comprehensive analyses will be presented in subsequent manuscripts currently in progress.

I would feel reassured by your comments, as an editor, that you do not agree with the reviewer's statement that preliminary results do not belong in scientific results. We are considering whether we should mention that this is the first issue instead of calling it preliminary.

Additionally, I do not understand why the reviewer mistrusts the SPC. It is a well-established blowing snow sensor that outputs both particle size and numbers, representing transport flux every second. Many researchers in the field of blowing snow have used this system as the most reliable sensor, not only in wind tunnels but also under harsh conditions like the alpine regions of the Alps and even in Antarctica (e.g., Naaim et al. 2010 and 2014, Wever et al. 2023). It has also been applied in sand transport research as a "Sand Particle Counter." The reviewer suggests that the signal is not a genuine feature of snow flux. However, snow flux over the snow surface is not uniform, so it is natural that the intensity shown in the figure changes with time and space, not due to sensor sensitivity. As is also explained in the previous reply, all the SPC sensors are properly calibrated with specific procedures before the observations; thus, the sensitivity and accuracy of all sensors are consistent. Over the snow surface, we can generally avoid the effect of dust and fine soil contamination over the optical parts, which differs from sand and soil surfaces. Numerous researchers overseas have attested to this and have kept measurements for long periods in the Alps and Antarctica. Generally, we can leave the system for an entire winter without

cleaning or wiping the optical parts. This campaign was carried out for less than two months, so we believe the contamination over the optics can be reasonably neglected.

The following description was added to the manuscript:

L69-77: Incidentally, it has also been applied in sand transport research as a "Sand Particle Counter" (Yamada et al., 2002 and Mikami et al., 2005). Accurate calibrations of all sensors are carried out beforehand, allowing us to reasonably exclude the effect of sensitivity differences. Over the snow surface, we can generally avoid the effect of dust and fine soil contamination over the optical parts, which differs from sand and soil surfaces. Numerous observations overseas have attested to this and have kept measurements for long periods in the Alps (e.g., Naaim-Bouvet et al., 2010, 2014, and Gilbert, 2019), the Arctic (e.g., Lenaerts et al., 2014 and Frey et al., 2020), and Antarctica (e.g., Sigmond et al., 2021 and Wever et al., 2023). Generally, we can leave the system for an entire winter without cleaning or wiping the optical parts. This campaign was carried out for less than two months, so we believe the contamination over the optics can be reasonably neglected.

As pointed out by the reviewer two, the 1.5-meter spacing in our measurements may not be fine enough. However, increasing the number of sensors narrows the gap between them, which can substantially affect both the air and blowing snow particle flow. This presents a dilemma and requires a compromise. Smaller sensors, such as FlowCapt for snow (Trouvilliez et al., 2015) and Sensit for sand (Stockton and Gillette, 1990), could be alternatives. However, it is well known that neither can obtain particle flux precisely (e.g., Lehning et al. 2002). Although the resolution may not always be fine enough to discern precise structures, our data analysis succeeded in drawing the approximate outline and structures of the streamer in Fig. 7. The 2D-autocorrelation of horizontal mass flux in Figure 8 indicates a lateral spacing of about 5 meters, which is more than three times larger than the sensor spacing of 1.5 meters. These explanations, including the limitations, are briefly described as follows:

L144-148:

Despite the 1.5-meter spacing between SPCs, which did not offer sufficient resolution for clarity in the figure, the analysis suggests that the structures at 1 cm above the surface had widths with the peak of flux around 30 cm wide and a lateral spacing of about 5 meters. The strong correlations are evident within plus/minus $\Delta Y= 5$ m in Fig. 8.

Although the reviewer two claims that Table 1 only reports quadrant percentages, not links with snow flux, it shows the percentage during the four blowing snow events. Furthermore, the correlation between the horizontal snow flux and the absolute value of the kinetic shear stress u'w' in each quadrant, and the relations between the snow flux, the fluctuating component of u' and w', and the product of u' and w' are shown in Figs 11 and 12 and discussed.

As mentioned above, hopefully the comments by both reviewers are reasonably satisfied. I hope you understand that the sheer volume of data obtained from our extensive observations presents a multitude of avenues for a more in-depth exploration of the spatiotemporal structures of blowing snow. Specific aspects, including the particle speed and wind speed in the proximity of these structures, as previously pointed out by the editor, are topics we are actively addressing in subsequent manuscripts currently in progress.

Lastly, following the suggestions from the editorial support team, we have checked the color schemes used in the figures with the Coblis Color Blindness Simulator. We have confirmed that readers with color vision deficiencies (Anomalous Trichromacy) will be able to recognize them without any difficulties.

Best regards,

Kouichi

**Dear Reviewer1,**

Thank you very much again for your careful review of our manuscript and for providing positive evaluations.

As is pointed out, we corrected the typo as follows:

Lines 137 and 305: from "Dynin" to "Dyunin."

**Dear Reviewer2,**

Thank you very much again for your careful review of our manuscript and for providing insightful comments, especially regarding the standpoint of sand particle transport. Although all of them are rather harsh review, I believe they are informative and educational to improve our manuscript.

First, the authors stress many times that this manuscript only presents "preliminary results", "a brief overview", and that "analysis is still quite limited". This is a very serious problem because scientific research publications should fundamentally not be preliminary, brief, or limited. Preliminary outputs may be presented at a conference, but they cannot be expected to become part of the established scientific literature, since any such preliminary work is by definition potentially still subject to change and modification. On this basis alone I believe this manuscript must be rejected because it is explicitly unable to contribute definitive scientific findings.

As I mentioned in my previous reply, this manuscript aims to present a concise overview of our unique observations and the obtained results, as a first issue. The dataset from our extensive observations is substantial, providing various avenues for concrete conclusions about the spatiotemporal structures of blowing snow. More comprehensive analyses will be presented in subsequent manuscripts currently in progress.

I would feel reassured by the editor, that he does not agree with the reviewer's statement that preliminary results do not belong in scientific results. We are considering whether we should mention that this is the first issue instead of calling it preliminary.

Second, my central concerns about the SPC data artefacts have been ignored. The authors stress that their SPC sensors are well calibrated and they give additional technical details, but they entirely fail to engage with my comments on the clearly artificial nature of the unrealistically

elevated/depressed signals for specific sensors over long periods of time, as visually evident still in the revised snow flux 'maps' of figures 4c, 7, 10a, and 13a. In figure 13a, for example, it is abundantly clear that the elevated signal at Y=3 over a period of 8 seconds is not a genuine feature of variable snow flux, and further scrutiny of the snow flux maps shows that most individual sensors produce signals that are consistently at different relative intensities. The authors have not addressed this issue, nor have they recognised the likelihood that sensor optics are fouled by the challenging field conditions (i.e. independent of calibrations) and the artefacts continue to affect their interpretations.

SPC is a well-established blowing snow sensor that outputs both particle size and numbers, representing transport flux every second. Many researchers in the field of blowing snow have used this system as the most reliable sensor, not only in wind tunnels but also under harsh conditions like the alpine regions of the Alps and even in Antarctica (e.g., Naaim et al. 2010 and 2014, Wever et al. 2023). It has also been applied in sand transport research as a "Sand Particle Counter." The reviewer suggests that the signal is not a genuine feature of snow flux. However, snow flux over the snow surface is not uniform, so it is natural that the intensity shown in the figure changes with time and space, not due to sensor sensitivity. As is also explained in the previous reply, all the SPC sensors are properly calibrated with specific procedures before the observations; thus, the sensitivity and accuracy of all sensors are consistent. Over the snow surface, we can generally avoid the effect of dust and fine soil contamination over the optical parts, which differs from sand and soil surfaces. Numerous researchers overseas have attested to this and have kept measurements for long periods in the Alps and Antarctica. Generally, we can leave the system for an entire winter without cleaning or wiping the optical parts. This campaign was carried out for less than two months, so we believe the contamination over the optics can be reasonably neglected.

The following description was added to the manuscript:

L69-77: Incidentally, it has also been applied in sand transport research as a "Sand Particle Counter" (Yamada et al., 2002 and Mikami et al., 2005). Accurate calibrations of all sensors are carried out beforehand, allowing us to reasonably exclude the effect of sensitivity differences. Over the snow surface, we can generally avoid the effect of dust and fine soil contamination over the optical parts, which differs from sand and soil surfaces. Numerous observations overseas have attested to this and have kept measurements for long periods in the Alps (e.g., Naaim-Bouvet et al., 2010, 2014, and Gilbert, 2019), the Arctic (e.g., Lenaerts et al., 2014 and Frey et al., 2020),

and Antarctica (e.g., Sigmond et al., 2021 and Wever et al., 2023). Generally, we can leave the system for an entire winter without cleaning or wiping the optical parts. This campaign was carried out for less than two months, so we believe the contamination over the optics can be reasonably neglected.

Third, the author-response and the revisions in the manuscript still lack any recognition of the very significant limitations imposed by the 1.5 m lateral spacing of the instruments, i.e. that the spatial resolution is fundamentally limited to this scale and the data cannot, by definition, reveal any structures or trends that are smaller than this. The author-response claims that "…by adjusting the range of the color bar in the figures, the width of the streamers became reasonably thinner,…" but this is simply a visualization gimmick (further exacerbated by the relative sensor distortions); streamers are generally on the order of 10-20 cm wide, and the instrumentation array is fundamentally incapable of resolving the sort of spatial patterning (streamers) visible in figure 6b. Text in the conclusion (L214-216) claiming to have observed streamer families and nested streamers is simply not sustainable. It also means that the (new) 2D auto-correlation map of Figure 8 has a fundamental lateral (spatial) resolution of 1.5 metres and hence nearly all of the autocorrelation structure is simply an interpolation feature.

In actual, the 1.5-meter spacing in our measurements may not be fine enough. However, increasing the number of sensors narrows the gap between them, which can substantially affect both the air and blowing snow particle flow. This presents a dilemma and requires a compromise. Smaller sensors, such as FlowCapt for snow (Trouvilliez et al., 2015) and Sensit for sand (Stockton and Gillette, 1990), could be alternatives. However, it is well known that neither can obtain particle flux precisely (e.g., Lehning et al. 2002). Although the resolution may not always be fine enough to discern precise structures, our data analysis succeeded in drawing the approximate outline and structures of the streamer in Fig. 7. The 2D-autocorrelation of horizontal mass flux in Figure 8 indicates a lateral spacing of about 5 meters, which is more than three times larger than the sensor spacing of 1.5 meters. These explanations, including the limitations, are briefly described as follows:

L144-148:

Despite the 1.5-meter spacing between SPCs, which did not offer sufficient resolution for clarity in the figure, the analysis suggests that the structures at 1 cm above the surface had widths with

the peak of flux around 30 cm wide and a lateral spacing of about 5 meters. The strong correlations are evident within plus/minus $\Delta Y = 5$ m in Fig. 8.

Fourth, although the author response notes that "analysis was conducted with all the sensor data," this is only true for the spatio-temporal maps of snow flux and wind forcing, but instead these are severely limited because only very short snippets of time (on the order of 10s of seconds) are shown and discussed. Other results, such as the auto-correlation, power spectra, and quadrant analysis, are severely limited because they are derived only from single instruments or from just one specific measurement run. There is insufficient consistent analysis of all available data as well as its internal variability to produce any reliable findings.

We believe it is not always necessary to present all data, as this manuscript is not a data report. For example, in this study, we introduced typical two-minute data segments from three different days. Similar strategies are also found in sand transport studies, such as Baas and Sherman (2006). While the analysis presented here is limited to specific cases, the amount of data obtained in this study is enormous. Analyzing all the available data is not practical and not always meaningful.

We believe that analyzing several typical cases, selected methodically rather than arbitrarily, and using these analyses to start discussions is a standard approach for reaching conclusions. In fact, we analyzed five cases, not just one, to discuss the relationship between wind speeds and mass fluxes with the parametric curves in Fig. 14.

Partly due to the above problems, the conclusions are still essentially trivial; the only finding that can be somewhat substantiated is the notion that snow flux correlates better with u'>0, but this is also already well-known from the sand transport literature (as is recognised here too). Even this is still quite limited though because the relevant results were only obtained from one measurement run. Authors claim that Table 1 also supports this finding, but this is not correct as that table only reports quadrant percentages, not links with snow flux.

Although the reviewer claims that Table 1 only reports quadrant percentages, not links with snow flux, it shows the percentage during the four blowing snow events. Furthermore, the correlation between the horizontal snow flux and the absolute value of the kinetic shear stress u'w' in each quadrant, and the relations between the snow flux, the fluctuating component of u' and w', and the product of u' and w' are shown in Figs 11 and 12 and discussed.

---

## Author Response (AR3)

**Dear Editor,**

Thank you very much indeed for giving the useful suggestions and comments.

We corrected the typo, the definite article, grammatical errors and so on, following your comments. Further, your suggestions are taken into account and we amended the manuscript.

**Editor: Please consider adding a few words to describe how this calibration is performed**

Following description was added to explain the calibration procedure.

All SPC sensors are accurately calibrated in advance using spinning wires of various diameters, enabling us to effectively account for any sensitivity differences. Detailed procedures are provided in Sato et al. (1993).

**Editor: Unclear sentence: please consider rephrasing.**

We changed the description as follows.

Numerous long-term observations have been conducted in the Alps (e.g., Naaim-Bouvet et al., 2010, 2014; Gilbert, 2019), the Arctic (e.g., Lenaerts et al., 2014; Frey et al., 2020), and Antarctica (e.g., Sigmond et al., 2021; Wever et al., 2023), all of which attest to the reliability of the SPC.

**Editor: What is meant by "clarity in the figure". Consider rephrasing as, e.g., "which limits the spatial resolution",**

**I am not sure to understand how a 30cm-peak can be identified with a 1.5m spatial resolution. Please consider elaborating on the influence of the interpolation procedure on this result and on the autocorrelation shown in Fig. 8. You could maybe**

**consider adding a lower limit on the spatial resolutions that are attainable?**

Unfortunately, it is hard to describe the resolution specifically. Instead, we added the explanation of the interpolation procedures applied in this analysis. The Delaunay triangulation is a strong By minimizing skinny triangles, it ensures that the interpolation is more stable and accurate and is widely used in various applications, including interpolation, mesh generation, and computer graphics.

Hope it will be of help.

The 1.5-meter spacing between SPCs may not always be sufficient to capture precise structures. However, the Delaunay triangulation (Cheng et al., 2012) used in Figure 7 is a powerful tool for interpolation, mesh generation, and graphical applications. It is widely used in Geographic Information Systems (GIS) to create terrain models, where triangulating elevation points constructs a surface that accurately represents the terrain with minimal distortion. Consequently, the structures observed at 1 cm above the surface, with widths around 2 meters, peak fluxes about 30 cm wide, and lateral spacing of approximately 5 meters, as shown in Figure 7, are quite plausible. Notably, the 2D autocorrelation of horizontal mass flux in Figure 8 indicates a lateral spacing of about 5 meters, which is more than three times the 1.5-meter sensor spacing.